# Forest fragmentation impacts the seasonality of Amazonian evergreen canopies

Matheus Henrique Nunes [1✉], José Luís Campana Camargo [2], Grégoire Vincent[3], Kim Calders [4], Rafael S. Oliveira [5], Alfredo Huete[6], Yhasmin Mendes de Moura[7,8], Bruce Nelson [9], Marielle N. Smith[10], Scott C. Stark[10] & Eduardo Eiji Maeda [1,11]

Predictions of the magnitude and timing of leaf phenology in Amazonian forests remain highly controversial. Here, we use terrestrial LiDAR surveys every two weeks spanning wet and dry seasons in Central Amazonia to show that plant phenology varies strongly across vertical strata in old-growth forests, but is sensitive to disturbances arising from forest fragmentation. In combination with continuous microclimate measurements, we find that when maximum daily temperatures reached 35 °C in the latter part of the dry season, the upper canopy of large trees in undisturbed forests lost plant material. In contrast, the understory greened up with increased light availability driven by the upper canopy loss, alongside increases in solar radiation, even during periods of drier soil and atmospheric conditions. However, persistently high temperatures in forest edges exacerbated the upper canopy losses of large trees throughout the dry season, whereas the understory in these light-rich environments was less dependent on the altered upper canopy structure. Our findings reveal a strong influence of edge effects on phenological controls in wet forests of Central Amazonia.

[1] Department of Geosciences and Geography, University of Helsinki, Helsinki 00014, Finland. [2] Biological Dynamics of Forest Fragment Project, National Institute for Amazonian Research, Manaus, AM 69067-375, Brazil. [3] AMAP, Univ Montpellier, IRD, CIRAD, CNRS, INRAE, Montpellier, France. [4] CAVElab—Computational and Applied Vegetation Ecology, Department of Environment, Faculty of Bioscience Engineering, Ghent University, Ghent, Belgium. [5] Department of Plant Biology, Institute of Biology, University of Campinas, Campinas, Brazil. [6] School of Life Sciences, Faculty of Science, University of Technology Sydney, Sydney, NSW 2007, Australia. [7] Institute of Geography and Geoecology, Karlsruhe Institute of Technology (KIT), Kaiserstr. 12, 76131 Karlsruhe, Germany. [8] Centre for Landscape and Climate Research, School of Geography, Geology and the Environment, University of Leicester, Leicester LE17RH, UK. [9] National Institute of Amazonian Research, Manaus, Brazil. [10] Department of Forestry, Michigan State University, East Lansing, MI, USA. [11] Area of Ecology and Biodiversity, School of Biological Sciences, Faculty of Science, The University of Hong Kong, Hong Kong, Hong Kong SAR. ✉email: matheus.nunes@helsinki.fi

L eaf phenology of Amazonian forests is a key component controlling the exchange of energy and trace gases—water vapour, carbon dioxide and volatile organic compounds— with influences on vegetation feedbacks on regional and global climates[1–5]. In the past decade, several studies have demonstrated from field data and remote sensing that a majority of Amazonian forests respond to climatic variations[2,6]. There is also mounting evidence that evergreen canopies exhibit seasonal variations[7–11] with changes in leaf demography and canopy structure[12]. Long-term studies have shown that 60–70% of species of humid Amazonian forests flush new leaves in the dry months[12,13] linked to higher solar radiation[4,14], which leads to increases in gross primary productivity as a result of new young leaves with higher photosynthetic capacity and water-use efficiency[4,15,16]. However, when some Amazonian forests are impacted by water stress, leaf development is reduced[17] and trees shed their leaves[10,18], with significant effects on leaf area dynamics[19]. To complicate matters further, leaf phenology also responds to multiple genetic factors, which have evolved to maximise photosynthetic and water-use efficiency during the dry season, reduce plant competition for light and water, and minimise herbivore pressure[7,16,20–22].

The impacts of climatic variations on leaf phenology can also be exacerbated by forest fragmentation[23]. Forest edges contain a large abundance of early successional species with resource-acquisitive strategies that maximise new leaf production and growth[24,25], but may be more vulnerable to climatic variations[26]. Forest fragmentation can increase the evaporative demand due to higher temperatures and wind exposure, and soil moisture can be lower at fragment edges[27], which may cause leaves to drop and lead to higher branch turnover[12,23]. However, ground observations of litterfall in Amazonian forests have shown only mild seasonality near edges[28]. Indeed, substantial uncertainty remains regarding the responses of fragmented forests to climatic seasonality, particularly because drought resistance varies among species[29–31] and surviving trees may acclimate or be adapted to the drier, hotter conditions near edges[32]. As the number of contiguously forested areas is significantly decreasing in the Amazon[33], understanding the effects of forest fragmentation on phenology is crucial for predicting alterations to canopy function in fragmented forests.

Seasonal variations in leaf quantity and leaf area across evergreen Amazonian forests have frequently been considered negligible or small[4,12,21,34]. However, these studies are based on passive optical remote sensing approaches, which cannot detect potential differences between canopy strata. These approaches tend to detect only upper canopy trees with deeper roots and water access[30], and that are likely adapted to more stressful conditions such as high solar radiation, high temperatures and low air humidity[35]. Active remote sensing observations from LiDAR may provide new insights into the interacting biophysical factors controlling phenology since LiDAR pulses penetrate the vertical canopy. Repeated terrestrial laser scanning (TLS, or 'terrestrial LiDAR') measurements can monitor subtle changes in forest structure[36], and provide observations of the balance between new leaf development (flush of new leaves, plant growth) and loss to abscission (leaf and branch fall) that can be separated across forest strata. Furthermore, the detailed and precise structural measurements offered by this system can help answer fundamental questions about the three-dimensional (3D) ecology of trees[37] without suffering from potentially confounding artefacts present in passive optical satellite images[11,34]. Recently, LiDAR-based studies have shown that leaf phenology in Amazonian forests is stratified over canopy positions, with understory growth occurring when abscission in the upper canopy contributes to increased light penetration in the lower canopy layers[19,38].

Here, we investigate the phenology of forests within the Biological Dynamics of Forest Fragments Project (BDFFP) in Central Amazonia, the world's longest-running experimental study of habitat fragmentation[39]. We use terrestrial laser scanning (TLS, or 'terrestrial LiDAR') surveys collected every 15 days spanning the wet and dry seasons to investigate how forest fragmentation and microclimatic seasonality interact to affect plant area of the understory and the upper canopy. Using a combination of 11 repeat TLS surveys, as well as continuous air temperature and soil moisture measurements in undisturbed old-growth forests and fragmented forests under edge effects, we hypothesised that: (1) vertically stratified plant phenology in undisturbed forests varies with seasonal microclimatic conditions; (2) the understory phenology is dependent on seasonal variations in the upper layers of the canopy; and (3) plant phenology is sensitive to disturbances arising from forest fragmentation, with the hotter and drier conditions of edges exacerbating leaf and branch losses during the dry season. To our knowledge, the work presented in this paper is the first to analyse tropical forest phenology with high spatial resolution 3D measurements (Fig. 1) and the first to experimentally demonstrate the effects of forest fragmentation on the seasonal variation of leaf area, and its vertical stratification, combined with microclimate measurements.

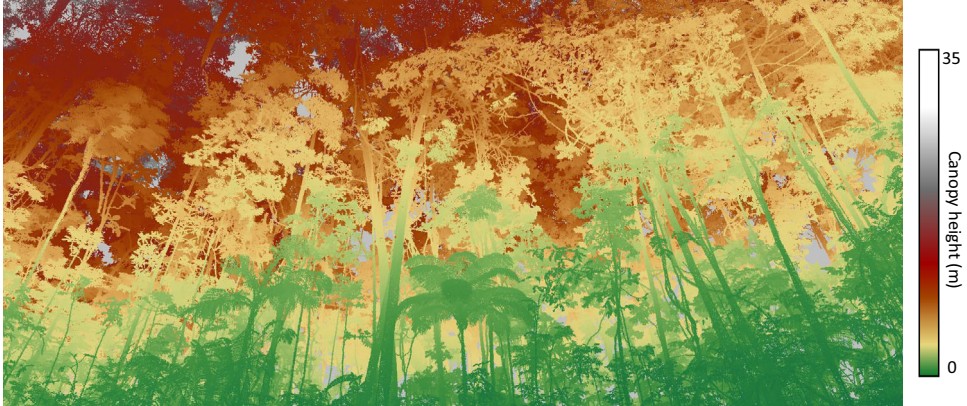

**Fig. 1 A view from a Central Amazonian forest understory.** Colours depict plants within distinct vertical strata. The high-speed terrestrial laser scanning (TLS) data acquisition of 500,000 measurements per second provides detailed measurements capable of detecting fine-scale changes in vegetation structure. We used a scan resolution of 40 mdeg in both azimuth and zenith directions, which results in a point spacing of 1.4 cm at a 20 m distance from the scanner. The laser pulse repetition rate used was 600 kHz, allowing a measurement range of up to 350 m and up to eight returns per pulse.

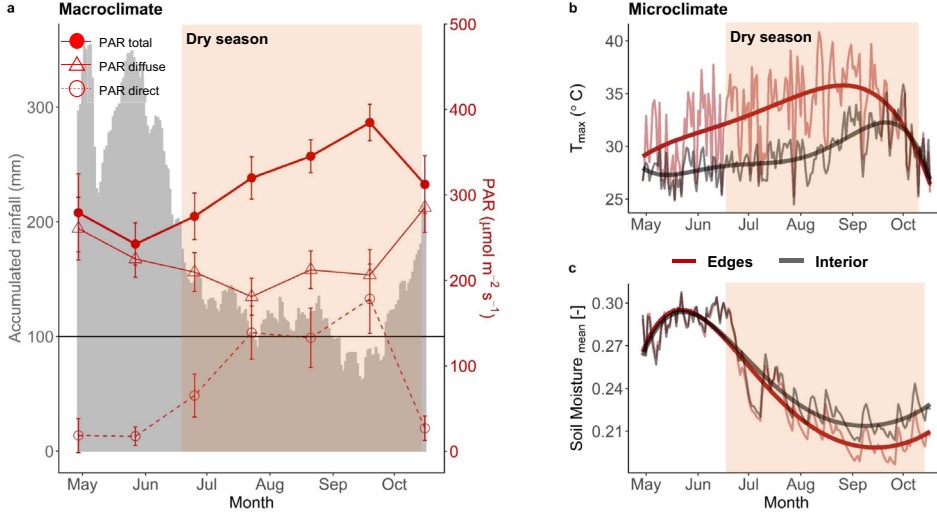

**Fig. 2 Climatic seasonality in Central Amazonian forests. a** Estimated diffuse, direct and total (diffuse + direct) Photosynthetic Active Radiation (PAR, in red) estimated from MODIS and accumulated 30 days (monthly) rainfall from NASA POWER (grey area). Each red point represents the monthly average calculated from daily estimates with error bars representing the 95% confidence intervals. **b** Maximum daily temperatures in the edge understory (red) and interior (dark grey) and **c** mean daily soil moisture (as volumetric water soil content: cm³ water/cm³ soil) measured continuously every 15 min in the edges (red) and in the interior of fragments (dark grey). Fifth order polynomial models were fit to the microclimate data for visualisation purposes. The shaded area corresponds to the dry season, defined as the period with accumulated monthly precipitation <200 mm month⁻¹.

## Results

**Seasonal climatic trends in Central Amazonian forests.** Daily precipitation estimates indicate the occurrence of a 4-month period of accumulated rainfall below 200 mm month$^{-1}$, and significant reductions in soil moisture between July and September in Central Amazonia (hereinafter referred to as "dry season"). The term "dry season" indicates a period of lower water availability and does not necessarily indicate that forests were water-limited. This dry season was coincident with a period of high MODIS-estimated Photosynthetic Active Radiation (PAR, Fig. 2a) and with significant increases in the locally measured understory temperature of interior forests and fragment edges (Fig. 2b, c). Forest fragmentation led to higher local temperatures in the edges, while water moisture in the soil remained unaffected by edge effects.

**Seasonal PAI variation and fragmentation effects.** Repeated TLS data acquired in two transects of $100 \times 10$ m and one transect of $30 \times 10$ m between April and October 2019 every 15 days (except between the end of April and early June when the duration between measurements was 40 days) were used to calculate Plant Area Density (PAD, one-sided area of plant material per unit of volume in m² m$^{-3}$). PAD is a combination of the leaf area and the surface area of woody components, including branches and trunks. An analysis of the vertical profile of the vegetation revealed the existence of only two vertical axes of variation during the dry season, with positive PAD changes below the height of 15 m above the ground (referred to as understory) and negative PAD changes above 15 m height (referred to as upper canopy) (Supplementary Fig. 3). The sum of PADs for each 1 m² vertical column ($X$-, $Y$-coordinate) was then calculated to obtain PAI, one-sided area of plant material per unit of ground surface in m² m$^{-2}$ (Fig. 3a). Nonlinear mixed models demonstrated that distance from forest edges has significant effects on PAI within 35–40 m of forest margins (Supplementary Fig. 2), and we, therefore, considered edge in this study categorically as the forests within 40 m of the forest fragment margins and interior as the forests at least 40 m distant from the fragment margins. These results demonstrate the existence of vertical and

horizontal within-season trends in phenology that should be considered when analysing across-season trends.

The TLS time-series revealed a strong vertical variability in the timing and magnitude of seasonal changes in the PAI of both structurally undisturbed forests and forests under edge effects. While transects exhibited similar phenological trends, PAI differed significantly between them (Supplementary Fig. 4). We then used linear mixed models to detect the effects of edges on the seasonality of understory PAI, upper canopy PAI, and total PAI (understory + upper canopy PAI), whilst controlling for spatial effects caused by transect differences by including edge effects nested within transect identity as random effects. The most parsimonious model (based on AIC; see Supplementary Table 1) to predict PAI for both the understory and total PAI was Eq. (1) which includes the additive effects of season and edge effects on PAI, both as categorical variables, and their interactive effects. Eq. (2) was selected for upper canopy PAI, which includes the effects of edge and an interaction term between edge effects and season. (Supplementary Table 1; Fig. 3; Supplementary Fig. 5).

$$PAI_i = \beta_0 + \beta_1 \, time_i + \beta_2 \, edge \, effects_i + \beta_3 \, time_i \times edge \, effects_i + u_i + \varepsilon_i$$

(1)

$$PAI_i = \beta_0 + \beta_1 \, edge \, effects_i + \beta_2 \, time_i \times edge \, effects_i + u_i + \varepsilon_i,$$

(2)

where $PAI_i$ is the plant area index in transect i, $\beta_0$ and $\beta_{1-3}$ are the fixed effect parameters, $u_i$ is the random intercept for edge effects nested within transect i (1 | Transects/Edge effects), and $\varepsilon_i$ is residual error. Both time and edge effects (edges versus interior) were treated as categorical variables.

In forest interiors, losses in the understory preceded the dry season, while significant losses in the upper canopy occurred at the end of the dry season (Fig. 3c, Supplementary Fig. 5b and 5d). More specifically, the PAI of the understory declined rapidly between April and early June ($t = -3.4$; $P$-value < 0.001) and reached a 5.3% ($-0.43$ m² m$^{-2}$) decline by late July ($t = -4.2$; $P$-value < 0.001). The PAI then increased to a full recovery ($+0.43$ m² m$^{-2}$) in September ($t = -1.2$; $P$-value = 0.21). By contrast, the upper canopy layer

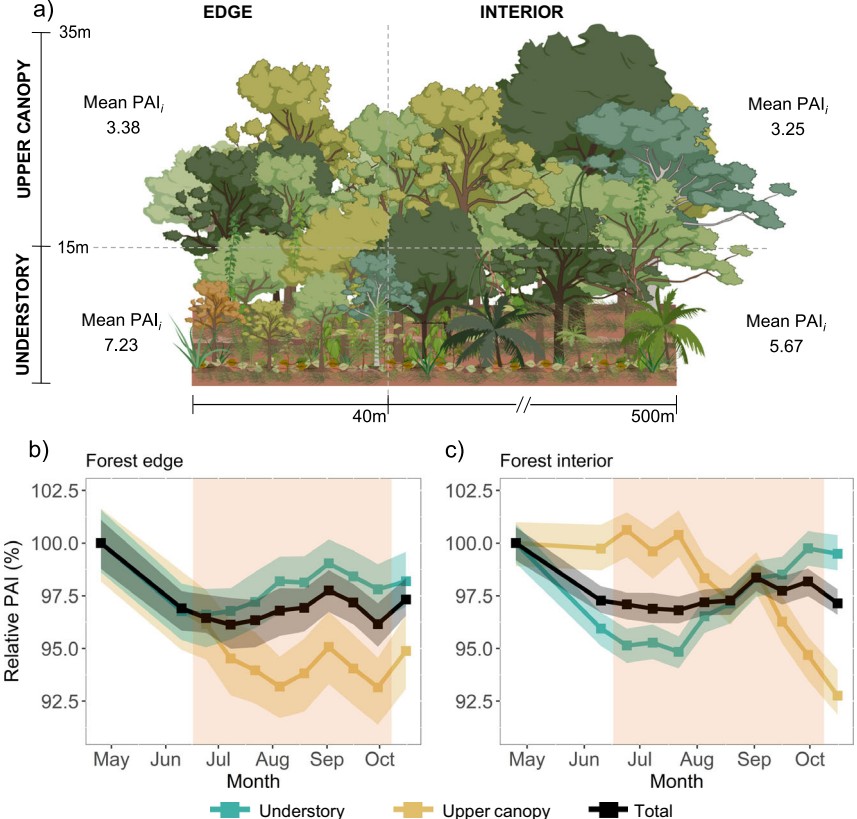

**Fig. 3 Predicted relative Plant area index (PAI, %) time-series. a** Forest phenology acquired using a Terrestrial Laser Scanner (TLS) within the Biological Dynamics of Forest Fragments Project (BDFFP) was vertically stratified, with the understory (<15 m aboveground) and upper canopy (≥15 m aboveground) presenting different trajectories of growth during the dry season. However, the vegetation structure and phenology were both significantly altered by edge effects up to 40 m from the forest fragment margins. PAI predictions from linear mixed modelling used measurement date (time), a categorical variable that indicates whether plots were near an edge (edge effects) and an interaction term time × edge effects as fixed variables. Edge effects nested within transect identity were included as random variables (Eqs. 1 and 2). Predicted relative PAI of the understory, upper canopy and total PAI (that combined both vertical strata) in **b** forest edges and **c** undisturbed interior forests was calculated as the PAI at any time divided by initial PAI (PAI$_i$) collected in April 2019. Each point (and lines corresponding to linear interpolations between points) represents the mean relative PAI obtained by fitting 200 randomised permutations of subsets split into 80/20 for calibration and validation, respectively. The shaded areas represent 95% confidence intervals based on uncertainty in those parameter estimates. While transects exhibited similar phenological trends, PAI differed significantly between transects—thereby we here show the predicted values. See Supplementary Fig. 4 for measured PAI and Supplementary Fig. 5 for absolute predicted PAI values and model uncertainty. The shaded area represents the dry season between mid-June and mid-October.

showed an inverse seasonal pattern to the understory; the upper canopy PAI remained relatively stable from April–September, but experienced a 7.6% ($-0.25\,\mathrm{m^2\,m^{-2}}$) decrease in late September ($t = -3.9$; $P$-value < 0.001).

The PAI time-series of forest edges (Fig. 3b, Supplementary Fig. 5a and 5c) showed significantly distinct patterns in comparison with those observed in forests distant from edges (indicated by the significant time × edge effects interaction term; Supplementary Table 1). Despite a subtle decline in PAI between April and July of 3.4% ($-0.08\,\mathrm{m^2\,m^{-2}}$) (Fig. 3b), the PAI in the edge understory did not show significant seasonal changes ($t = -1.7$; $P$-value = 0.07). However, the upper canopy of edges had significant PAI losses of 6% ($-0.25\,\mathrm{m^2\,m^{-2}}$) by mid-July ($t = 2.2$; $P$-value < 0.05), nearly 3 months before the upper canopy of interior forests was significantly affected (Fig. 3c).

The temporal patterns of total PAI in forest edges and forest interior had patterns that reflected the combination of the stratified phenological trends. In the forest interior, a decrease of 2.7% ($-0.34\,\mathrm{m^2\,m^{-2}}$) was observed between April and early July ($t = -2.8$; $P$-value < 0.005), and remained relatively stable throughout the dry season. The phenology of forest edges showed very similar trends to forest interior when distinct strata were not

considered; the total PAI decreased by 3.2% ($-0.25\,\mathrm{m^2\,m^{-2}}$) between April and early July ($t = -2.2$; $P$-value = 0.03), and also remained relatively stable throughout the dry season. These results show that when the seasonal patterns are not vertically stratified, the PAI trends for the edges versus interiors are strikingly similar and mainly driven by the understory PAI, where the majority of the plant area is.

**Seasonal variations in forest microclimate and vertically stratified PAI.** We also illustrate the significant seasonal variations in PAI against the microclimatic conditions measured in the edges of the fragment and in the forest interior (Supplementary Fig. 6 and 7). Losses in PAI of canopies in edges and forest interior occurred when temperatures were elevated (above 35 °C; Supplementary Fig. 6d). Losses in PAI of upper canopies in the forest edges preceded canopy losses in the interior by 3 months, which coincided with temperatures 3–5 °C hotter in edges throughout the dry season than interior environments (Supplementary Fig. 6c); this strongly supports the idea that the seasonal dynamics of Amazonian forests at the upper canopy level is dependent on temperature, and that fragmentation exacerbates

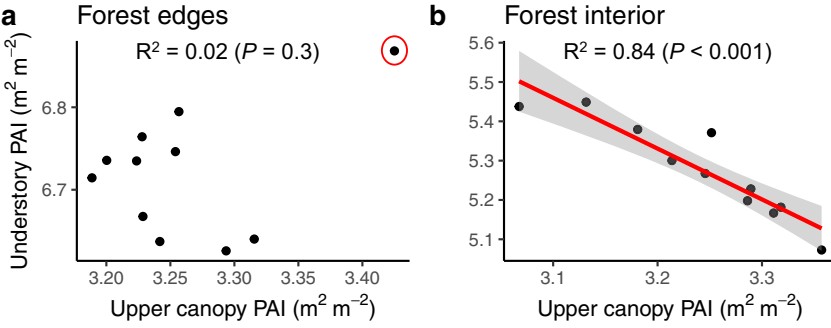

**Fig. 4 Observed seasonal changes in upper canopy PAI (Plant Area Index, $m^2 m^{-2}$) versus understory PAI.** LiDAR-based PAI data measured between April and October 2019 in Central Amazonian forests were classified as **a** forest edges, situated within 40 m from the forest fragment margins, and as **b** forest interior, situated at least 40 m from the fragment margins. Black dots represent the mean value of each Terrestrial Laser Scanning (TLS) survey based on 3480 understory and 3480 upper canopy PAI values in the forest interior and 1653 understory and 1653 upper canopy PAI values in the forest edges. Model's $R^2$ and $P$-value were calculated from simple linear regression (Understory PAI = $\beta_0 + \beta_1$ Upper canopy PAI). The red line in panel b represents predicted values by the model, with the shaded grey area corresponding to the two-sided 95% confidence intervals. The highlighted point in red in panel (**a**) denotes the first TLS measurement made in April 2019. We excluded this point to further investigate the covariance between strata but found no significant relationship (Supplementary Fig. 8).

these effects. On the other hand, the understory of interior forests had sharp decreases in PAI between April and June, a period when soil moisture was still high, and maximum temperatures were relatively low (27.8 ± 0.6 °C, Supplementary Fig. 7b and 7d). However, the understory PAI of these forests increased with losses in the upper canopy PAI, even with increases in temperature and high solar radiation (Fig. 2a), and a full recovery in plant area occurring when the temperatures peaked in September. These findings support the idea that light is a more important control of the forest understory than temperature and water availability.

**Relationships between understory and upper canopy seasonal variations**. We then investigate whether decreasing leaf area in the understory of forest edges and in the forest interior was synchronised with variation in upper canopy plant area. We found a strongly negative linear correlation between variations in PAI of the upper canopy and understory in the forest interior only (F-value = 54.4; P-value < 0.001; $R^2$ = 0.84; Fig. 4). There was no relationship in edges (F-value = 1.2; P-value = 0.29; $R^2$ = 0.02), which aligns with our hypothesis that fragmentation changes phenological patterns; in this case, affecting the seasonal pattern of understory dependency on upper canopy plant area.

## Discussion

Repeat high density terrestrial LiDAR combined with microclimate measurements of a Central Amazonian forest provided a unique perspective on the seasonal dynamics of vegetation and the interaction with fragmentation. PAI—as leaf area index (LAI) and the surface area of woody components—showed inverse patterns in the understory versus upper canopy. In the structurally undisturbed interior of a large forest fragment, plant area in the understory decreased by ~5% before the start of the dry season and fully recovered by mid dry season in September. Conversely, the upper canopy (>15 m aboveground) of interior forest maintained its canopy structure throughout most of the dry season, with the greatest losses (~8%) in upper canopy PAI occurring from September to mid-October, when the microclimate reaches its lowest soil moisture and maximum temperatures are high (above 35 °C). Variations in plant area in the understory were strongly coordinated with upper canopy changes in PAI ($R^2$ = 84%, Fig. 4), which suggests that leaf flush in the understory follows increasing light availability as plant area is lost

in the upper canopy. Edge effects, however, changed the phenological patterns observed in interior forests; while edge effects exacerbated upper canopy loss throughout the dry season, the understory was less seasonal. The pattern of higher leaf loss in the upper canopy, where large trees dominate, is consistent with edge effects enhancing leaf stress and creating periods of high evaporative demand—indeed, temperatures were consistently 3–5 °C higher and soil moisture levels lower in the dry season in the forest edges. This study demonstrates the value of repeated terrestrial LiDAR surveys, which allow the detection of fine-scale changes in forests without potential artefacts of passive remote sensing studies[36], and provide a perspective on forest dynamics and its spatial variability that is difficult to achieve with lower resolution remote sensing approaches.

Our PAI time-series of interior forests indicated upper canopy losses that are sensitive to elevated temperatures, whereas the understory maintains high leaf production under high light availability mediated by upper canopy dynamics, even during periods of drier soil and atmospheric conditions during non-El Niño years. Passive remote sensing and field observations have demonstrated that Central Amazonian forests "green up" during the dry season[9,11], but with negligible increases in PAI[4,12]. Our findings demonstrate, instead, stratified canopy responses to seasonally mediated environmental conditions and suggest that large trees may mediate a green up in the lower canopy. We show that if phenological patterns are not vertically stratified, total canopy PAI (the combined understory and upper canopy PAI) tends to reflect the understory PAI (where most of the PAI is). These results suggest that if differences between strata are not considered alongside changes in LAI, litterfall production and leaf demography, predictions of the climatic influences on vegetation may be undermined or misleading.

Vertical differences in phenology may arise from a direct response to changing light availability in the understory and from contrasting functional and hydraulic properties between canopy and understory trees. Recent studies in Amazonian forests have shown that leaf area increases in the understory occur under maximal irradiance conditions when the upper canopy layer is partially deciduous during the dry season[38,40], as diffuse and direct solar radiation in the understory can increase linearly with decreasing upper canopy plant area[41]. The dominant species in the understory of Amazonian forests are distinct from upper canopy dominants and are differentiated and more complex in functional strategies[35,42]. Understory trees have xylem that is

more embolism resistant and can tolerate more negative water potentials in the dry season without risking hydraulic failure compared to upper canopy trees, which tend to be more vulnerable to drought-induced embolism[30]. High embolism resistance of understory trees allows an anisohydric stomatal behaviour (low degree of regulation) and the maintenance of high stomatal conductance at the peak of the dry season[30]. The high drought tolerance of understory trees is also likely to be a key trait allowing them to flush new leaves during periods of water stress. In contrast, canopy trees exhibit lower embolism resistance, high stomatal sensitivity and significant declines in photosynthesis during periods of high atmospheric demand and low soil water availability[43]. The loss of upper canopy leaves in Amazonian forests at the end of the dry season is consistent with the importance of water availability for leaf development[38], and suggests that canopy trees in these forests may be vulnerable to periods of high evaporative demand[2,22]. These results challenge the paradigm of Amazonian large trees being necessarily capable of accessing deep water and hence being primarily light-limited[44,45].

This study presents a dataset of fine-scale, high-frequency LiDAR, elucidating the magnitude and timing of forest phenology and impact of fragmentation from one of the most important experiments on tropical forest fragmentation (BDFFP). However, the generality of our findings across years and sites, particularly across large-scale Amazonian gradients in seasonality, edaphic properties, and soil moisture, remains to be tested. At a site in the eastern central Amazon with trees attaining a maximum size ~10 m taller (Tapajós National Forest, Pará[46]), Smith and colleagues found more complex vertically and horizontally stratified dynamics[19]. Here, in contrast to our study, the upper canopy above 20 m increased in plant area towards the end of the dry season. On the other hand, the sunlit canopy surface zone below 20 m—common in this vertically heterogeneous site with a large amount of canopy gaps—decreased through the late dry season, consistent with our upper canopy results. The shaded understory layer below 20 m increased as the mid-canopy decreased, displaying a strong anticorrelation pattern that could be analogous to the forest interior understory vs upper canopy relationship that we observed. The canopy surface near 20 m in the Tapajós could be more functionally analogous to the upper canopy of the BDFFP where, indeed, much of the dry season decrease in plant area occurred between 15 and 25 m height (Supplementary Fig. 3). However, what explains the contrast in the 'above 20 m' Tapajós upper canopy? These sites differ in fertility, mean rainfall, length of dry season, and other factors[47]. Upper canopy trees in the Tapajós access deep soil water[30], while more typically wet forests near Manaus may not root as deeply, potentially because of smaller tree sizes[48] or functional selection[49]. Overall, these results suggest that there may be environmental (including belowground factors) and species-specific trait-linked controls on canopy structure and phenological climate responses. Better understanding these controls can help us predict variation in climate response across the Amazon.

Ground measurements are preferred to study subtle changes in canopy density in comparison to scans from above the canopy. Ground measurements are immune to seasonal changes in vegetation-ground reflectance ratios that may affect the transmittance estimated from airborne and spaceborne LiDAR systems, and which may otherwise negatively affect LAI estimates[10]. Such artefacts may have contributed to the negative correlations between upper canopy and understory LAI from the Amazonian-scale IceSat-based estimates presented in Tang and Dubayah[38,40]. Although our sampling effort attempted to minimise uncertainties in the PAI estimates, we did not account for (i) changes in leaf orientation and light transmittance caused by leaf age and

changes in plant water content[50,51], or (ii) a potential bias induced by not separating wood from leaves in the estimation of PAI, as leaf turnover rates can be different from branch turnover rates[52]. However, both (i) and (ii) require an automated separation of leaves from woody materials, which may contribute to additional uncertainties that can vary through space and time[53].

We observed strong edge effects that changed the phenological patterns observed in the interior forests. Upper canopy PAI losses were significantly affected by the dry season in forest edges, occurring nearly 3 months before upper canopy losses in the interior forests. Dry season temperatures in forest edges were 3–5 °C higher than in the interior of the fragment, while changes in soil moisture were small. These higher temperatures may lead to an increase in vapour pressure deficit (VPD), inducing stomatal closure and leaf loss[23,43,54,55], as shedding leaves may help to avoid the desiccating effects of water and heat stress[56]. On the other hand, plant area in the understory of forest edges was unaffected by the seasonal microclimatic changes or losses in upper canopy leaf area. The aseasonality of plant area in the understory of edges indicates that leaf production rates were similar to leaf loss rates during wet and dry seasons. These edges are dominated by pioneer species[25] that thrive under the light-rich environment caused by lateral light penetration and by the formation of gaps associated with the mortality of large trees[57]. These conditions may disrupt the between strata light-mediated anticorrelation of leaf area dynamics since edge understories are less affected by variations in the upper canopy structure.

This study sheds light on the seasonal trends in the plant areas of Amazonian forests and highlights complex interacting effects of climate and human disturbance on forest phenology. The total leaf quantity (LAI) is a key component modulating tree growth[58] and net primary productivity[59]. The consistently higher and aseasonal understory PAI in these fragment edges may explain the increased growth rates of understory tree species in these edges in comparison to the same tree species in interior forests[60]. However, the dry season losses in upper canopy plant area near the edges 3 months earlier compared with interior forests likely represent a shortening of the photosynthetic-active period of large trees, potentially reducing photosynthetic carbon fixation (gross primary productivity; GPP). If $CO_2$ uptake of the upper canopy is suppressed, this may have negative consequences for investment in tissue maintenance and defence[61], which may, in turn, increase the mortality of large trees that dominate upper canopies and contribute to a large reduction in the aboveground biomass of these forests[62]. Carbon losses from forest degradation already exceed those from deforestation in the Amazon[63], and fragmentation is a large contributor to degradation-associated carbon emissions[64]. Given the drier and warmer future projected for the Central and Eastern regions of the Amazon, and extended dry-season length[65], our findings suggest that fragmentation will exacerbate the negative effects of high temperatures on the upper canopy of these forests. Considering that fragment edges cover a total area of 176,555 km$^2$ of Amazonian forests[66], the thermal sensitivity of canopies on the edges of fragmented forests could translate into a large component of edge-related carbon losses.

Predicting changes in phenology is particularly challenging, given that the timing of biological events results from an interaction of organism functional traits, genetic background and environmental factors[67]. Much progress has been made to understand the seasonality of Amazonian tree species and communities at local and regional levels[4,7,9,11,14,31,38]. Nonetheless, our results show that the variability in phenology that arises from canopy stratification and edge effects has large impacts on plant area seasonality. The lack of edge effects on the seasonal variance of total plant area highlights the challenge faced by passive sensors onboard satellite platforms; these systems may suffer from a

flattened perspective with data strongly influenced by canopy layers with a denser plant area and little ability to detect significant height-stratified forest canopy responses to climate. Efforts to separate plants occupying different strata and habitats are needed to address this challenge, which is aligned with recent debates on the effects of strata on regional patterns of species dominance and composition in Amazonian forests[35]. Unoccupied aerial vehicle-borne laser scanning should be instrumental in this respect. These systems have the capacity to collect high density point clouds at a high temporal frequency over relatively large areas (up to tens of ha) and offers the opportunity to characterise leafing patterns at the scale of individual crowns[68]. At landscape and regional scales, airborne and satellite-based active LiDAR sensors can also provide a crucial height-stratified perspective (e.g., NASA's new GEDI mission)[69,70].

Despite our progress characterizing height and environmentally stratified canopy phenology, the mechanisms that control phenology at the species level remain elusive. Changes in PAI may not capture the leaf exchange dynamics as it is unknown what proportion of species and trees shed their leaves completely prior to flushing new leaves, and those that go through a more progressive leaf exchange. A mixture of the above strategies can produce a stable PAI even in case of strong seasonality in leaf exchange patterns. We propose that future research on phenology should continue to untangle the interactions of the environment with functional and phylogenetic diversity both within and among species. TLS can be particularly useful in this context; tree segmentation allows for 3D architectural reconstruction and the calculation of structural metrics[37,71]. TLS-based phenological data at the tree and species levels can help elucidate mechanisms controlling phenology in the Amazon such as (i) the specific environmental factors determining phenology, (ii) the molecular and physiological processes regulating phenology, and (iii) whether variation in phenology reflects genetic differences (high interspecific variation) or plastic responses to environmental heterogeneity (high within-species variation). This may help resolve outstanding debates concerning the mechanisms by which species respond to seasonal climatic variations and improve predictions of plant responses to global changes.

## Methods

The study was conducted in Central Amazonian forests (2°20 30 ′S, 60° 05 37 W) within the Biological Dynamics of Forest Fragments Project (BDFFP), the world's longest-running experimental study of habitat fragmentation[39]. The region has seen notable carbon and biodiversity losses due to forest fragmentation effects[25,66] and is predicted to be markedly impacted by climatic changes[72]. The pioneering BDFFP project sites are composed of forest fragments originally isolated in 1980 by converting mature forest into cattle pastures. Currently, the 'matrix' between the forest fragments is dominated by secondary growth forests, but a 100 m strip surrounding the forest fragments is cleaned regularly by cutting vegetation regrowth to maintain their isolation (Supplementary Fig. 1a). As an experimental control that minimises additional anthropogenic influences such as illegal logging, hunting, fire penetration and pollution, the project offers insights into ecological and environmental changes in fragmented forests. We selected a 100-ha forest fragment to investigate phenological responses within transects varying in distance from the fragment edges (0–500 m). At the community level, the forest edges of our study are dominated by a high density of early successional, fast-growth species, because of the elevated tree mortality near forest edges and seed dispersion from degraded neighbouring habitats, while the centre of the fragment comprises undisturbed primary forest[25,39].

### Terrestrial laser scanning: data acquisition, registration and PAI estimation.
The TLS data were acquired using a RIEGL VZ-400i system between April and October 2019 every 15 days, except between the end of April and early June when the duration between measurements was 40 days (we clarify in the analysis section how we addressed artefacts attributed to sampling effort). We used a scan resolution of 40 mdeg in both azimuth and zenith directions, which results in a point spacing of 34 mm at 50 m distance from the scanner. The laser pulse repetition rate used was 600 kHz, allowing a measurement range of up to 350 m and up to eight returns per pulse. The scans covered two transects of 100 × 10 m near the fragment edges and perpendicular to the forest fragment margins measured 11 times and one

transect of 30 × 10 m length in the centre of the forest fragment measured ten times. The transect in the centre lies 500 m from any fragment margin to ensure sampling of forest interior in the absence of edge effects on the canopy structure (~40 m[73]). This sampling strategy covered a total area of 0.52 ha, which included 274 trees with diameter at breast height (DBH) ≥ 10 cm, lianas, shrubs, saplings, seedlings and acaulescent palms that were repeatedly measured 11 times.

To ensure a full 3D representation of the upper canopy (35 m in height), each transect consisted of three scan lines parallel to each other with scans spaced at 5 m intervals within and between lines (Supplementary Fig. 1b). The distance between scanning positions was smaller than the 10–40 m usually applied in previous studies to minimise data uncertainties due to occlusion in dense tropical forests and maximise data acquisition in the upper canopy[74]. Given that the RIEGL VZ-400i has a zenith angle range of 30–130°, an additional scan was acquired at each sampling location with the scanner tilted at 90° from the vertical position. A total of 276 scans across all transects each time resulted in a complete sampling of the full hemisphere in each scan location (Supplementary Fig. 1c). All scans were later co-registered into a single point cloud per transect using the RiSCAN PRO software version 2.9, provided by RIEGL. Given that the RIEGL VZ-400i uses onboard sensor data with an algorithm to align scans without the use of reflectors, automatic registration was done before a final adjustment of scans.

We used the LAStools (rapidlasso, GmbH; Gilching, Germany) suite of computational tools to process the data. To minimise errors in the fusion of the repeated scans, we first created a common digital terrain model (DTM) at 0.5 m resolution using a combination of ground returns from the first survey. Using an inverse distance weighting algorithm in the function grid_terrain from the "lidR" package in R, a common DTM was constructed from LiDAR ground returns. Plant area density (PAD) for all transects was then calculated using a voxel-based approach (with a 5 m buffer around each transect to maximise the PAD data). The volume occupied by vegetation within each transect was divided into 1 $m^3$ voxels, and the PAD calculated for each of these voxels (Supplementary Fig. 1d). This procedure was done in the LiDAR data voxelization software AMAPVox[75,76]. AMAPvox tracks every laser pulse through a 3D grid (voxelised space) to the last recorded hit. The effective sampling area of each laser pulse (or fraction of pulse in case of multiple hits) is computed from the theoretical beam section (a function of distance to laser and divergence of laser beam) and the remaining beam fraction entering a voxel. In case more than one hit is recorded for a given pulse, the beam section is equally distributed between the different hits of the pulse. This information is combined with the optical path length of each pulse entering a voxel to compute the local transmittance or equivalently the local attenuation per voxel. Different estimation procedures are provided in the AMAPVox software. We used the Free Path Length estimator first developed for single return TLS in Pimont and colleagues[76] and later extended to the multiple return case[77]. The common assumption made for all estimation procedures in AMAPvox is to consider vegetation elements as randomly distributed within a voxel (thereby neglecting within voxel clumping) and to express the directional gap probability (or directional transmittance) as a function of the optical path length of laser pulse through a voxel and the local extinction coefficient[78]. The extinction coefficient is the product of the Plant Area Density and the projection function $G(\theta)$, which is the ratio of plant area projected in direction $\theta$ to actual area:

$$P(\theta, l) = \exp(-\lambda_\theta \times l), \qquad (3)$$

where $P(\theta,l)$ is the probability of non-interception of a light beam of zenith angle $\theta$ (i.e. directional gap probability) along a path of length l, $\lambda_\theta$ ($m^{-1}$) is the directional attenuation coefficient, and l is the optical path length (m).

The PAD ($m^2\ m^{-3}$) is related to $\lambda$ as follows:

$$PAD = \lambda_\theta / G(\theta) \qquad (4)$$

$G(\theta)$, the plant projection function, is taken equal to 0.5, assuming a spherical distribution of leaf inclination angles[79]. This function is likely to be spatially variable in complex forest canopies.

In total, the number of voxels was 230,609, which were monitored 11 times during seasonal changes. We then calculated the sum of PADs for each 1 $m^2$ vertical column (X-, Y-coordinate) to obtain the PAI, which is a combination of the leaf area index and the area of wood components, including branches and trunks.

Albeit restricted in their spatial extent, our densely sampled repeated terrestrial laser scans likely provide more accurate and robust measures of PAI than any other method previously used to monitor seasonal changes in plant areas of Amazonian forests. Measurement accuracy is enhanced by the extremely high sampling density and multiple return capacity of the laser system, and multiple view angles reducing the area occluded to the sensor[36]; fast-to-operate single return laser profilers such as portable ground LiDAR as used by Smith and colleagues[19] while having high pulse density may have more limited accuracy[80].

### Determining edge effects and number of forest strata.
To test the hypotheses that (1) fragmentation has significant effects on the structure of the vegetation in the BDFFP experiment—following Almeida and colleagues[73]—and (2) that edge effects impact phenology, we related PAI collected during the 11 TLS campaigns with distance from the edge using a nonlinear mixed model. The Eq. (5) included the term $\exp^{(-x)}$, as an asymptotic component that represents the saturation of PAI with distance from the edge, denoted by $x$ in the model, and transect as a random

variable, allowing us to include any idiosyncratic differences between transects.

$$PAI = \beta_0 + \beta_1 * \exp^{(-\beta_2 * x)} + u_i + \varepsilon_i \quad (5)$$

where $\beta_0$ to $\beta_2$ are the model parameters, $u_i$ is the random intercept for transect i, and $\varepsilon_i$ is the normally distributed residual error.

This approach has been used to investigate edge effects on forest structure and dynamics[23,81]. The model was fitted using the function nlme in R. The results from this model indicates that the effects of transect account for 6.5% of the total PAI variability only, and that most of the variance (93.5%) is explained by the within-transect variability, including the distance from edge and seasonal variations (Supplementary Fig. 2). A hockey-stick model consisting of two linear segments was also implemented with the R package "SiZer" using the function piecewise linear. This model identified a "distance from edge" threshold, dividing voxels from the two transects near the fragment margins into edge and interior groups. We demonstrate the edge effects on PAI within 37 m of forest margins (Supplementary Fig. 2). These results corroborate a previous study in the same forest fragments showing edge effects of up to 40 m on canopy height[73]. Therefore, we considered edge in this study as the forests within 40 m of the forest fragment margins, which resulted in two edge transects ($2 \times 40$ m) and three interior transects ($2 \times 60$ m + $1 \times 30$ m).

We also tested the hypothesis suggested by Smith and colleagues[19] that the lower and upper strata of the vegetation have asynchronous changes in plant area during the dry season by comparing PAD on October 16th with PAD on June 24th in these strata. Species, functional and phylogenetic composition of the understory are distinct from the upper canopy in Central Amazonian forests[30,35]. While the understory is comprised of lower branches, seedlings, shade-tolerant and embolism resistant trees and shrubs, lianas, acaulescent palms and saplings of young adult trees, the upper canopy is made up of adult predominantly shade-tolerant species, including tall and emergent trees and lianas. We then calculated the changes in PAD during the dry season to investigate shifts in the vertical profile of vegetation to elucidate the seasonal responses of specific strata (Supplementary Fig. 3a, 3b).

We observed consistent positive PAD changes below the height of 15 m above the ground and negative PAD changes above 15 m height (Supplementary Fig. 3b). Thus, given the existence of only two axes of variation along the vertical profile of the vegetation, we utilized this height to define understory (<15 m aboveground) and upper canopy (≥15 m aboveground) in this study. This is consistent with a prior study in the Amazonian forest, which also demonstrated distinct seasonal responses in leaf area above and below a height of 15 m[19]. The sum of all the understory PADs and the upper canopy PADs are referred to as understory PAI and upper canopy PAI, respectively. Our analysis comprises 5133 PAI values for the understory and 5133 PAI values for the upper canopy, each monitored 11 times during the seasonal climatic variations. The understory accounts for 62 + 1.1% of the total PAI in the forest interior and 68 + 0.4% of the total PAI of forest edges throughout the period of measurement (Supplementary Fig. 3a).

**Climatic variables to elucidate the timing in PAI seasonal changes**. PAI changes may be controlled by changes in micro and macroclimatic conditions[19,38,40]. We demonstrate below how we estimated solar radiation and accumulated rainfall at the landscape level, and continuously measured air temperature and soil moisture in the understory of forest edges and interior of forest fragments to examine the synchrony between these factors and the PAI time-series in the understory and canopy.

**Solar radiation and accumulated rainfall**. Leaf flushing in Central Amazonian forests coincides with peaks in PAR (W/m$^2$) during periods of low rainfall[4,14,38]. PAR varies significantly within forest canopies and changes over time due to variations in the incident solar flux density and solar direction[41]. Incident solar PAR contains two components: direct PAR and diffuse PAR—and the latter is mostly controlled by scattering of particles and cloud cover in the atmosphere[82]. The photosynthetic efficiencies of direct and diffuse PAR are different in forests, with positive effects of diffuse light on photosynthetic rates[83] and atmospheric $CO_2$ assimilation[84] in comparison to plants under direct light conditions. To examine the synchrony between PAR and seasonal PAI changes, we derived solar radiation from the product MCD18A2 V6 (https://lpdaac.usgs.gov/products/mcd18a2v006/). This product uses the bands of the visible spectrum (400–700 nm) of both sensors (Terra and Aqua) from the Moderate Resolution Imaging Spectroradiometer (MODIS) to estimate daily PAR at a 5-km pixel resolution[85]. Daily mean rainfall estimates were also derived from NASA's POWER (Prediction of Worldwide Energy Resources) data with a spatial resolution of 0.5° latitude by 0.5° longitude ($55 \times 55$ km). Meteorological parameters are derived from NASA's GMAO MERRA-2 assimilation model (https://gmao.gsfc.nasa.gov/reanalysis/MERRA/) and GEOS FP-IT (https://gmao.gsfc.nasa.gov/news/geos_system_news/2016/FP-IT_NRT_G5.12.4.php). We then integrated the daily rainfall estimates to accumulated monthly (30 day period) rainfall and classified dry season as the period with running 30-day rainfall below 200 mm as in Maeda and colleagues[86].

**Microclimate variables**. Soil moisture and maximum temperatures are key drivers of species' distributions and affect how species respond to climatic variations[87,88]. We measured air temperature (°C) and electrical conductivity of soil moisture (time-domain transmission; TDT) across a network of 22 data loggers varying in distance from the forest fragment margins (0 and 520 m). Temperature-Moisture-Sensor (TMS) data loggers measured air temperature at 15 cm above the ground and TDT at 8 cm below ground, and all the data retrieved using TMS Lolly manager software (Tomst, Czech Republic)[89]. TDT values were transformed into volumetric soil moisture following calibration curves in Wild and colleagues[89] using as input data soil texture (50% clay, 25% sand and 25% silt contents) and mean soil density of 1100 kg/m$^3$ measured by Camargo and Kapos[90] in the same forest fragments of our study. Data loggers were shielded from direct solar radiation and recorded data every 15 min. Microclimate data were recorded between 27th April 2019 and 16th October 2019, resulting in a total of 435,798 coupled temperature and volumetric soil moisture readings. TMS device measures microclimate variables affecting many ecological processes, including those related to water and energy balance. We calculated mean daily soil moisture and maximum daily soil moisture to investigate their synchrony with the PAI time series.

**Phenology modelling for interior and edge forests**. We used a linear mixed-effects (LME) model of understory PAI, upper canopy PAI and a combination of both strata (total PAI) measured from TLS in each transect as a function of time of measurement (time). We also included an interaction term with the plot category of location near an edge or in the forest fragment interior (edge effects) following Qie and colleagues[81]. The time × edge effects interaction represents how edge effects caused by forest fragmentation influence the seasonal variation in PAI. We compared this LME model with other LME models that contained the variables time and edge effects as additive terms to examine the significance of seasonality and fragmentation on PAI variation. Model explanatory power was assessed in terms of AIC (Supplementary Table 1). The LME model was fitted using the lme function in the "nlme" R package. Variations in transect area and monitoring period can influence PAI trends, and thus we used varIdent weights function to account for the noise attributed to sampling effort[91]. Performance of the final models was evaluated using an 80/20 split of the data for calibration and validation, respectively, over 200 randomised permutations of the dataset. These analyses generated a distribution of model coefficients and allowed an assessment of model stability and uncertainty of predictions. We calculated 95% confidence intervals from the 2.5% and 97.5% quantiles of the distribution of model coefficients.

If increasing upper canopy PAI contributes to lower light interception in the lower stratum of the vegetation, we may expect a decreasing leaf development in the understory of forests in the interior of fragments[38,41]. However, we may also expect that such an effect on understory PAI by increasing upper canopy PAI is reduced or absent near fragment edges, with the loss of tall trees and lateral light from forest edges exposing the understory to more direct sunlight[92]. We tested this by averaging the community-level PAI in understory and upper canopy strata for each census, and then using linear models (lm function in R) to examine the relationships of PAI between the understory and upper canopy.

**Reporting summary**. Further information on research design is available in the Nature Research Reporting Summary linked to this article.

## Data availability
The repeated PAI data collected in Central Amazonia using a terrestrial laser scanner (TLS) between April and October 2019 and generated in this study have been deposited in the national Finnish Fairdata services database under accession code https://etsin.fairdata.fi/dataset/e488f81b-b927-4bbd-a6f7-2f532f434b2b. PAR estimates were obtained from https://lpdaac.usgs.gov/products/mcd18a2v006/ and meteorological data from https://gmao.gsfc.nasa.gov/reanalysis/MERRA/. Microclimate measurements collected in the field during the current study are available from the corresponding author on reasonable request within 10 days.

## Code availability
There is no particular code or mathematical algorithm that is considered crucial to the conclusions. All relevant R-functions that were used are referred to in the Method section (see package vignettes for details).

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

## Acknowledgements

This study was funded by the Academy of Finland (decision numbers 318252, 319905 and 345472). This publication is number 831 of the Technical Series of the Biological Dynamics of Forest Fragment Forest (BDFFP—INPA / STRI). We thank the Biological Dynamics of Forest Fragment Project team for the thorough logistical support in the field. We are grateful to Renann Henrique Paiva Dias Silva, Juliane Menezes and Vinicius Bertin for the great and thorough assistance in the field. G.V. received support from Laboratoire d'Excellence CEBA (ANR-10-LABX-25). K.C. was funded by the European Union's Horizon 2020 research and innovation programme under the Marie Sklodowska-Curie grant agreement No 835398. R.S.O. receives a CNPq productivity scholarship and is supported by a NERC-FAPESP grant 2019/07773-1. Y.M.d.M. was supported by the Royal Society under the Newton International Fellowship funding (NF170036) and HPC-Europa-3 (HPC17TA3RL), supported by the H2020-European Commission. S.C.S. received support from the US NSF (DEB-1950080 and 1754357) and USDA NIFA.

## Author contributions

M.H.N. and E.E.M. conceived the study. M.H.N. and E.E.M. led data collection in the field. M.H.N., G.V. and E.E.M. processed the LiDAR data. M.H.N. performed data analyses and wrote the manuscript. M.H.N., J.L.C.C., G.V., K.C., R.S.O., A.H., Y.M.d.M., B.N., M.N.S., S.C.S. and E.E.M. contributed to the revision of the paper.

## Competing interests

The authors declare no competing interests.
