## [Peer Review File · Nature Communications]

Reviewer comments, first round review:

Reviewer #1 (Remarks to the Author):

This study examines the effects of fragmentation on the canopy and understory plant area index over time as a measure of phenology. The authors found that when it gets hot late in the dry season, the upper canopy of large trees in undisturbed forests shed leaves and branches, and the understory greens-up with the increased light availability. However, forest edges experience persistently high temperatures which causes canopy losses throughout the dry season, and the understory phenology is "disrupted".

The authors conclude that "forest fragmentation will aggravate forest loss under a hotter and drier future scenario." However, the implications of these findings seem overstated or at least not clearly supported. The connection between the phenology changes and "forest loss" is not at all clear.

I would also question the generality of these findings for a couple reasons. As the authors discuss, sites with different soil fertility or other conditions may exhibit quite different responses, so this study of 1 site (3 transects) in 1 year should be interpreted cautiously in terms of its applicability across years and other sites. Also, the mechanism of the observed differences remains unknown, as the different phenology could be driven by altered species composition in fragments or abiotic conditions, or both.

Other comments

The methods and analysis took a lot of effort to try to reconstruct what was done. Could use some more detail in the main text to understand what was done

There were 2 transects in fragmented forest; 1 in interior? But also analyzed "distance from edge" for the two transects on the edge. How is the data aggregated for Figures in main text categorized "edge" and "interior"? It's also unclear what the error bars represent in Fig 3.

The "disrupted" understory phenology seems to refer to a slightly less seasonal pattern in edges (Fig1), but the PAI in the edges were consistently higher than interior anyway (note different axis scales in Fig 1). The understory is really doing better in edges, in terms of biomass at least (although it could be different species, we don't know). So "disrupted" just means a changed phenological pattern, but it's not clear that this is a negative outcome for species or ecosystem processes.

L29 define "strata"

Follows with "this seasonality" – what's that referring to?

L74-75: in what way has previous work shown phenology is "stratified" over canopy positions and conditions? Can you be more explicit to set up the expectations? What have previous studies found?

Fig 3 – could condense to 1 graph instead of 4; would be easier to compare patterns

Fig 4a. appears to be totally driven by outlier.

Reviewer #2 (Remarks to the Author):

Overall this is a very interesting paper. A unique dataset, ideal for answering this question, has been collected, and some good analysis performed. I definitely think it should be published after some clarification.

My main criticism is that as it is presented, the author uses terms that have not yet been explained in enough detail for the reader to understand, and so follow the argument. In particular, line 126: This is the first mention of models or numbers in the main paper. It would be helpful to the reader to define what modelling you are doing and what the numbers mean before referring to them within the main paper. This sentence cannot easily be understood without that extra information. You have not yet stated that you intend to predict PAI with a model before this point in the paper?

I also got a bit confused as to which graphs showed observations estimated from the TLS and which showed outputs from your TLS and climate data models? In particular; are figure 3 modelled or measured results? If modelled, as said in the caption, why did you show the model outputs rather than show the measurements you took with the TLS? Or are you referring to TLS to PAI voxel model? You also use a model to relate to climate. Are the two models the same? Please make it clear which model is being used where, and if you did use a model with the climate data, why did you chose to do that rather than analyse the PAD/I estimated from the TLS. A bit more clarification here would be helpful.

I think that it is important to clarify that point above before this paper can be published.

Is it true to say that the total PAI is seasonally invariant? Looking at figure 3, if you add the overstorey and understorey there would be a large spike in May, with a much higher PAI than all other months. You show this in supplementary figure 4. What tests did you use to see if there was a seasonal variation?

On a similar subject: Line 129: "Note that the lack of time effect in Eq. 4 indicated that there is significant temporal variation in only the vertical distribution of PAI " - from the supplementary material, figure 4 shows that there is a change over time. Do the error bars here show uncertainty in the model or spatial variation? If the latter, I do not think it is correct to use that spread as a measure of the significance of the change over time. Please clarify?

Line 287: You have shown a link between edges and leaf loss, but does this necessarily translate into a carbon loss? Could you provide either data or a citation linking a reduction in mean PAI during the wet season with carbon loss to support that claim?

Figure 4's line fit for the edges seems to be dominated by the one outlier on the far right. Without that one point it looks as if there would have been a negative trend (although not as significant as the overstorey trend). Is that one point certain, or could there have been a mistake or significant occlusion affecting it?

Line 214 onwards is an interesting discussion, but is it directly relevant to the experiment here and is it supported by the data? If so I think it might be better in the literature review rather than within the discussion?

I think that supplementary material section 2 needs some more detail. Why are you using a model to relate distance from edge to PAI (or is the model a line fit to suppress noise)? Why not show the raw data? If you do need to use a model, I think you need to justify why it is needed and show its accuracy. That you are using a model suggests that the relationship is very noisy? If so, how significant is the relationship?

Supplementary section 4 needs more detail for a reader to fully understand and appreciate your points.

Supplementary materials line 82: "Each point represents the mean values predicted by mixed

modelling, with the error bars depicting the bootstrapped 95% confidence intervals." - I do not understand why you are using a model here? You have a direct measure of PAI from the TLS. Why not investigate that rather than passing it through a model, adding in assumptions and errors? A bit more clarification on exactly what analysis you did would be helpful.

Reviewer #3 (Remarks to the Author):

Overall comment:

In studying the influence of fragmentation on tropical forest phenology and density of vertical biomass, the authors investigate a potentially significant phenomenon with implications for biome-wide estimates of ecosystem function. The topic is of particular concern given the prevalence of forest edges across the Amazon.

The authors examine seasonal changes in the vertical biomass profile (i.e. understory vs. overstory) of the canopy, which as they point out can be difficult if not impossible to accomplish via passive spaceborne observations. This is due to the fact that the green biomass signal is integrated vertically and therefore cannot be decomposed into understory vs. overstory components in passive remote sensing data. The authors utilize terrestrial laser scanning in order to resolve these two canopy components through a series of field observations. The dataset is impressive with observations collected every 15 days to capture within- and across-season trends in phenology.

As a scientist with much experience in terrestrial laser scanning I find the remote sensing methods (data collection, processing, and analysis) to be solid. The results show small yet statistically significant trends in the PAI data; of particular note is the low level of PAI variation (due to the fact that ~90-97% of the canopy is invariant through time—comprised of woody and non-deciduous foliar elements). While this might be seen as a limitation to this study (i.e. that the lidar data do not allow resolution of foliar elements as well as some passive spectral approaches), I find the large size and novelty of the dataset in characterizing detailed PAI data is quite attractive and represents an important advance in our understanding of tropical forest phenology relative to disturbance.

Detailed Comments:

Lines 85-88: It would add clarity to the paper and help with understanding the specific scientific goals of the paper if the authors could frame their objectives using hypotheses. This is particularly necessary because three times later in the paper the authors allude to hypotheses (lines 182, 422, and 436) that were never clearly stated, and this is the place to do it.

Figure 1: Please edit caption to say "...depicting plants from distinct vertical strata." Also to help the reader interpret the scan image, please include a scale bar indicating the height of vegetation relative to the colors shown on the graphic.

Figure 2, line 118: Suggest: "...Fifth order polynomial models were fit to the microclimate data for visualization purposes..."

Equations, lines 132-133: Unless I'm missing a formatting protocol regarding the Methods, these should be labeled Eq.'s 1 and 2.

Figure 3: This is an interesting figure however I feel that it can be much improved. I suggest that the authors use a consistent scale bar on the y-axis so that the trends can be compared and contrasted more readily (in fact plotting upper/lower canopy on one graph each would likely be even better). In addition, to assist with interpretation of the overall forest trends, I suggest that a third series of plots be added that shows the overall PAI of the forest canopy in the forest edges vs. forest interior. This can be added without adding space to the figure, as in my opinion the figure would be improved if there was only one graph each for edges and interior forest locations, with each graph having three lines (upper, lower, and total canopy).

Line 178: I don't believe you have leaf area data? Only plant area index, of which a very large component (~90-97%) apparently is comprised of non-deciduous plant material based on Figure 3.

Lines 223-224: Showing these total canopy PAI data would be helpful (i.e. as a third line on Figure

3) so that the reader can follow. Also--It would seem that since the upper/lower canopy PAI trend is not negatively correlated in the disturbed edge effect case, that in fact there may in this case be a total canopy PAI sensitivity to climatic seasonality—please explain.

Line 307: I suggest that you use an actual spatial indicator of scale here (e.g. "Landscape", "Regional", "Continental" rather than "satellite" scale, which is vague since satellite observations occur from local to global scales.

Line 393: "as follows:"

Line 397: This first sentence is duplicative as it has already been stated that voxels are one cubic meter in volume.

Lines 408-419: This section would be better placed in the Discussion section, in my opinion.

REVIEWER COMMENTS

Reviewer #1 (Remarks to the Author):

This study examines the effects of fragmentation on the canopy and understory plant area index over time as a measure of phenology. The authors found that when it gets hot late in the dry season, the upper canopy of large trees in undisturbed forests shed leaves and branches, and the understory greens-up with the increased light availability. However, forest edges experience persistently high temperatures which causes canopy losses throughout the dry season, and the understory phenology is “disrupted”.

Thank you very much for your comments on the manuscript. We have addressed your points and concerns below and made changes in the manuscript that include your considerations.

The authors conclude that “forest fragmentation will aggravate forest loss under a hotter and drier future scenario.” However, the implications of these findings seem overstated or at least not clearly supported. The connection between the phenology changes and “forest loss” is not at all clear.

Thank you for your comment. We agree the term “forest loss” is not clear and ambiguous. We have changed our final sentence in the abstract to:

“Here, we reveal a plant-climate interaction controlling the seasonality of wet forests in Central Amazonia and demonstrate a strong influence of edge effects on phenological controls.”

We have also included a text in the Discussion that suggests long-term impacts of fragmentation on forest seasonality:

“The total leaf quantity (LAI) is a key component modulating tree growth (Lambers and Oliveira, 2019) and net primary productivity (Reich, 2012). The consistently higher and aseasonal understory PAI in these fragment edges may explain the increased growth rates of understory tree species in these edges in comparison to the same tree species in interior forests (Albiero-Junior, 2021). However, the dry season losses in upper canopy plant area near the edges three months earlier compared with interior forests likely represent a shortening of the photosynthetic-active period of large trees, potentially reducing photosynthetic carbon fixation

(gross primary productivity; GPP). If CO₂ uptake of the upper canopy is suppressed, this may have negative consequences for investment in tissue maintenance and defence (Doughty et al., 2015), which may, in turn, increase the mortality of large trees that dominate upper canopies and contribute to a large reduction in the aboveground biomass of these forests (San-José et al., 2021).”

I would also question the generality of these findings for a couple reasons. As the authors discuss, sites with different soil fertility or other conditions may exhibit quite different responses, so this study of 1 site (3 transects) in 1 year should be interpreted cautiously in terms of its applicability across years and other sites.

Thank you for your comment and we agree that our findings should be interpreted cautiously across years and other sites. Nonetheless, we argue that given the limited availability of in-situ PAI data for such a lengthy period and the high frequency repetition throughout the wet and dry seasons, our data and analysis may elucidate a potentially significant phenomenon with implications for biome-wide estimates of ecosystem function. The topic is of particular concern given the prevalence of forest edges across the Amazon. We would like to highlight that our study was conducted in the longest-running experiment in the tropics established to investigate the effects of forest fragmentation on biological dynamics, thereby minimising additional anthropogenic influences such as illegal logging, hunting, fire penetration and pollution, the project offers. We would also like to highlight previous studies on leaf phenology/ ontogeny based on phenocams from flux towers viewing small areas and that have also interpreted their results from small areas (Wu et al., 2016a, 2016 b).

We have included the following opening sentence in the 4th paragraph of the Discussion to make it clear that phenology may be subject to variations arising from biotic (i.e. species-specific traits) and abiotic factors (i.e. soil, radiation, water availability), and therefore our results should be interpreted cautiously. “This study presents a unique dataset of fine-scale, high frequency LiDAR, elucidating the magnitude and timing of forest phenology and impact of fragmentation from one of the most important experiments on tropical forest fragmentation (BDFFP). However, the generality of our findings across years and sites, particularly across large-scale Amazonian gradients in seasonality, edaphic properties, and soil moisture, remains to be tested.”

We have also included the Figure 4 in the Supplementary Material. These graphs may help understand that even though there is large variation in PAI with transect identity that may arise from environmental conditions (i.e. soil and topography) and species composition, the transects exhibit similar vertically-stratified phenological patterns.

Supplementary Figure 4. Observed Plant Area Index (PAI, $m^2 m^{-2}$) time-series per transect in interior forests (at least 40 m away from forest edges) in Central Amazonia. Large spatial variation in PAI may be attributed to varying local environmental conditions and species composition. While transects exhibited similar (a) understory and (b) upper canopy phenological trends, PAI differed significantly between them. Each point denotes the mean PAI value per transect in each measurement time (with lines denoting linear interpolations between points and colours representing the transect identity). Error bars are the 95% confidence intervals.

Also, the mechanism of the observed differences remains unknown, as the different phenology could be driven by altered species composition in fragments or abiotic conditions, or both.

We agree that species - and functional traits - can explain differences in phenology within and across Amazonian forests. Our intention to include the 4th paragraph in the Discussion was to

bring the attention of phenology variability across other sites - aligned with the idea that environmental factors and species may help explain some of these differences. Related to your comment above, we have included in the 4th paragraph of the discussion that better understanding species-specific trait-linked controls may help us predict variation in climate response across the Amazon.

We have also included a text in the final paragraph of the Discussion that highlights why the mechanisms that control phenology at the species level remain elusive and we propose that future research on phenology should continue to untangle the interactions of the environment with functional and phylogenetic diversity both within and among species:

“Changes in PAI may not capture the leaf exchange dynamics as it is unknown what proportion of species and trees shed their leaves completely prior to flushing new leaves, and those that go through a more progressive leaf exchange. A mixture of the above strategies can produce a stable PAI even in case of strong seasonality in leaf exchange patterns.”

Other comments

The methods and analysis took a lot of effort to try to reconstruct what was done. Could use some more detail in the main text to understand what was done

Thank you for pointing out the missing details that help the reader better understand what was done. We lost some of the details when keeping the results and discussion presented before the Methods and analysis section. We have included one paragraph in the section “**Seasonal PAI variation and fragmentation effect**” that explains the details of the methods and analysis. These details are better explained in the Methods section of the manuscript.

“Repeated TLS data acquired in two transects of 100 x 10 m and one transect of 30 x 10 m between April and October 2019 every 15 days (except between the end of April and early June when the duration between measurements was 40 days) were used to calculate Plant Area Density (PAD, one-sided area of plant material per unit of volume in $\text{m}^2 \text{m}^{-3}$). PAD is a combination of the leaf area and the surface area of woody components, including branches and trunks. An analysis of the vertical profile of the vegetation revealed the existence of only two vertical axes of variation during the dry season, with positive PAD changes below the height of 15 m above the ground (referred to as understory) and negative PAD changes above 15 m height

(referred to as upper canopy) (Supplementary Figure 3). The sum of PADs for each 1 m² vertical column (X, Y coordinate) were then calculated to obtain Plant Area Index (PAI, one-sided area of plant material per unit of ground surface in m² m⁻²) (Figure 3a). Non-linear mixed models demonstrated that distance from forest edges has significant effects on PAI within 35 - 40 m of forest margins (Supplementary Figure 2), and we therefore considered edge in this study categorically as the forests within 40 m of the forest fragment margins and interior as the forests at least 40 m distant from the fragment margins. These results demonstrate the existence of vertical and horizontal within-season trends in phenology that should be considered when analysing across-season trends.

The TLS time-series revealed a strong vertical variability in the timing and magnitude of seasonal changes in the PAI of structurally undisturbed forests and forests under edge effects. While transects exhibited similar phenological trends, PAI differed significantly between them (Supplementary Figure 4). We then used linear mixed models to detect the effects of edges on the seasonality of understory PAI, upper canopy PAI, and total PAI (understory + upper canopy PAI), whilst controlling for spatial effects caused by transect differences by including edge effects nested within transect identity as random effects. The most parsimonious model (based on AIC; see Supplementary Table 1) to predict PAI for both the understory and total PAI was equation (1) which includes the additive effects of season and edge effects on PAI, both as categorical variables, and their interactive effects. Equation (2) was selected for upper canopy PAI, which includes the effects of edge and an interaction term between edge effects and season. (Supplementary Table 1; Figure 3; Supplementary Figure 5).”

There were 2 transects in fragmented forest; 1 in interior? But also analyzed “distance from edge” for the two transects on the edge. How is the data aggregated for Figures in main text categorized “edge” and “interior”? It’s also unclear what the error bars represent in Fig 3.

Thank you for your comment. The scans covered two transects of 100 x 10 m near the fragment edges and perpendicular to the forest fragment margins measured 11 times and 1 transect of 30 x 10 m length in the centre of the forest fragment measured 10 times. The transect in the centre lies 500 m from any fragment margin to ensure sampling of forest interior without edge effects (Laurance et al., 2006).

We first modelled the effects of edge effects on the canopy structure using nonlinear mixed modelling to identify a threshold with clear influences of edge effects on canopy structure. We included exp^{-x} as an asymptotic component that represents the saturation of PAI with distance from edge, denoted by x in the model, and transect as a random variable, allowing us to include any idiosyncratic differences between transects. This model identified a “distance from edge” threshold, dividing voxels from the two transects near the fragment margins into edge and interior groups (Supplementary Methods 2). We demonstrate the edge effects on PAI within ~ 40 m of forest margins (Supplementary Fig. 2).

We then model the influences of edge effects as categorical variables (edges versus interior). We have included figure 3a to help explain the design of the phenology experiment and included more explanation in the manuscript in the section “Determining edge effects and number of forest strata” in the Methods section. We also highlight in the Results section that our predicted PAI time-series uses time and edge effects as categorical variables.

We have also changed the error bars to confidence interval in Fig 3b and 3c as in “The shaded areas represent 95% confidence intervals based on uncertainty in those parameter estimates”.

The “disrupted” understory phenology seems to refer to a slightly less seasonal pattern in edges (Fig1), but the PAI in the edges were consistently higher than interior anyway (note different axis scales in Fig 1). The understory is really doing better in edges, in terms of biomass at least (although it could be different species, we don’t know). So “disrupted” just means a changed phenological pattern, but it’s not clear that this is a negative outcome for species or ecosystem processes.

Thank you for your comment. We agree with you that the understory has a consistently higher and aseasonal PAI. Indeed, studies from the same forest fragment edges have demonstrated that understory tree species can have higher growth rates in the edges in comparison to interior forests. We have included the following sentence in the 7th paragraph of the Discussion: “The consistently higher and aseasonal PAI may explain the increased growth rates of forests of understory species in these edges in comparison to the same species in interior forests (Albiero-Junior, 2021)”. We also have changed the word disrupted to changed or impacted throughout the text to make it more clear that we refer to the fragmentation effects on the canopy, including a less seasonal pattern in the understory and the effects of fragmentation on the between-strata

anticorrelation observed in the interior forests. Note that we no longer have different axis scales – we have changed to relative PAI (%) to make comparisons simpler.

L29 define “strata”

Follows with “this seasonality” – what’s that referring to?

Thank you. We have made it clear in the text as in “... plant phenology varies strongly across vertical strata in old-growth forests, but is sensitive to disturbances arising from forest fragmentation”

L74-75: in what way has previous work shown phenology is “stratified” over canopy positions and conditions? Can you be more explicit to set up the expectations? What have previous studies found?

Thank you for your comment. We clarify here and in the text what previous studies have found. “... leaf phenology in Amazonian forests is stratified over canopy positions, with understory growth occurring when abscission in the upper canopy contributes to increased light penetration in the lower canopy layers (Tang and Dubayah, 2017; Smith et al., 2019)”.

Fig 3 – could condense to 1 graph instead of 4; would be easier to compare patterns

Thank you for your comment. We have condensed to 2 graphs (Figure 3b and 3c) as we want to clearly visualise the phenological patterns in the forest edge versus interior. We have transformed the values to relative PAI (%) to make it easier to compare patterns between the edge and interior. We also summarise the initial PAI values (measured in April 2019) of understory and upper canopy strata at the edge and interior in Figure 3a.

However, we believe that showing the absolute predicted PAI data is also relevant as it gives a better picture of the magnitude of changes. We present the absolute predicted PAI values in the Supplementary Material (Supplementary Figure 5) - and each point in the graph represents the predicted PAI from a subset of the data (more information is included in the figure caption).

Figure 3. Predicted relative Plant Area Index (PAI, %) time-series. PAI predictions from linear mixed modelling used date of LiDAR measurements and the interaction with a categorical variable indicating whether plots were near an edge as fixed variables. Edge effects nested within transect identity were included as random variables (equations 1 and 2). Predicted PAI of the understory (< 15 m canopy height), upper canopy (> 15 m canopy height) and total PAI that combined both vertical strata in b) forest edges and c) undisturbed interior forests. Forest edges are defined as canopies within 40 m from forest margins while forest interior are canopies at least 40 m away from the forest fragment margins. Relative PAI was calculated as the PAI at any time divided by the initial PAI. Each point (and lines denoting linear interpolations between

points) represents the mean relative PAI obtained by fitting 200 randomised permutations of subsets split into 80/20 for calibration and validation, respectively. The shaded areas represent 95% confidence intervals based on uncertainty in those parameter estimates. While transects exhibited similar phenological trends, PAI differed significantly between transects (Supplementary Figure 4 for measured PAI). For absolute predicted PAI values and model uncertainty, see Supplementary Figure 5. The shaded grey area represents the dry season.

Fig 4a. appears to be totally driven by outlier.

Thank you for your interesting comment. We have included more information about the graph in the caption. Each black dot represents the mean from 3480 understory and 3480 upper canopy PAI values measured in each survey in the forest interior and 1653 understory and 1653 upper canopy PAI values measured in each survey in the forest edges. The measurements protocol (i.e. position of measurement, LiDAR specifications) remained constant during all periods to minimise analytical errors. To minimise the effect of uncertainties due to occlusion in these dense Amazonian forests, the distance between scanning positions was smaller than the 10-40 m usually applied in previous studies to minimize data uncertainties and maximize data acquisition in the upper canopy. Therefore, we believe that the understory versus upper canopy PAI relationship in the edges is not driven by the one outlier on the far right - as we controlled for sources of uncertainties in the PAI estimation and used a large dataset to obtain the mean value for each period of measurement.

To complement our explanation, we reran the understory versus upper canopy PAI in the edges as per the figure below and found that even after ignoring this specific point their co-variation is weak and not significant.

Figure 4. Observed seasonal changes in upper canopy PAI versus understory PAI. LiDAR-based Plant Area Index (PAI, $m^2 m^{-2}$) between April and October 2019 from Central Amazonian forests. Black dots represent the mean from 3480 understory and 3480 upper canopy PAI values measured in each survey in the forest interior and 1653 understory and 1653 upper canopy PAI values measured in each survey in the forest edges. The highlighted point in red in panel (a) denotes the first TLS measurement made in April 2019. We excluded this point to further investigate the covariance between strata but found no significant relationship (see Supplementary Figure 8). The red lines represent values predicted by simple linear regression (i.e. Understory PAI = $\beta_0 + \beta_1$ Upper canopy PAI), with the shaded grey area depicting the 95% confidence intervals.

Figure not included in the paper. **Observed seasonal changes in upper canopy PAI versus understory PAI.** This graph differs from Figure 4a as it ignores the highlighted point. We show that variation in understory PAI in edges does not co-vary with upper canopy PAI even after excluding the mean PAI value measured in April 2019.

REFERENCES

Albiero-Junior, A., Venegas-Gonzalez, A., Camargo, J.L.C., Roig, F.A. and Tomazello-Filho, M., 2021. Amazon forest fragmentation and edge effects temporarily favored understory and midstory tree growth. *Trees*, pp.1-10.

Doughty, C.E., Metcalfe, D.B., Girardin, C.A.J., Amezquita, F.F., Cabrera, D.G., Huasco, W.H., Silva-Espejo, J.E., Araujo-Murakami, A., Da Costa, M.C., Rocha, W. and Feldpausch, T.R., 2015. Drought impact on forest carbon dynamics and fluxes in Amazonia. *Nature*, 519(7541), pp.78-82.

Lambers, H. and Oliveira, R.S., 2019. Plant water relations. In *Plant physiological ecology* (pp. 187-263). Springer, Cham.

Laurance, W.F., Nascimento, H.E., Laurance, S.G., Andrade, A., Ribeiro, J.E., Giraldo, J.P., Lovejoy, T.E., Condit, R., Chave, J., Harms, K.E. and D'Angelo, S., 2006. Rapid decay of tree-community composition in Amazonian forest fragments. *Proceedings of the National Academy of Sciences*, 103(50), pp.19010-19014.

Reich, P.B., 2012. Key canopy traits drive forest productivity. *Proceedings of the Royal Society B: Biological Sciences*, 279(1736), pp.2128-2134.

San-José, M., Werden, L., Peterson, C.J., Oviedo-Brenes, F. and Zahawi, R.A., 2021. Large tree mortality leads to major aboveground biomass decline in a tropical forest reserve. *Oecologia*, pp.1-12.

Smith, M.N., Stark, S.C., Taylor, T.C., Ferreira, M.L., de Oliveira, E., Restrepo- Coupe, N., Chen, S., Woodcock, T., Dos Santos, D.B., Alves, L.F. and Figueira, M., 2019. Seasonal and drought- related changes in leaf area profiles depend on height and light environment in an Amazon forest. *New Phytologist*, 222(3), pp.1284-1297.

Tang, H. and Dubayah, R., 2017. Light-driven growth in Amazon evergreen forests explained by seasonal variations of vertical canopy structure. *Proceedings of the National Academy of Sciences*, 114(10), pp.2640-2644.

Wu, J., Albert, L.P., Lopes, A.P., Restrepo-Coupe, N., Hayek, M., Wiedemann, K.T., Guan, K., Stark, S.C., Christoffersen, B., Prohaska, N. and Tavares, J.V., 2016. Leaf development and demography explain photosynthetic seasonality in Amazon evergreen forests. *Science*, 351(6276), pp.972-976.

Wu, J., Serbin, S.P., Xu, X., Albert, L.P., Chen, M., Meng, R., Saleska, S.R. and Rogers, A., 2017. The phenology of leaf quality and its within- canopy variation is essential for accurate modeling of photosynthesis in tropical evergreen forests. *Global Change Biology*, 23(11), pp.4814-4827.

Reviewer #2 (Remarks to the Author):

Overall this is a very interesting paper. A unique dataset, ideal for answering this question, has been collected, and some good analysis performed. I definitely think it should be published after some clarification.

Thank you very much for reviewing the manuscript and for the comments you have provided. We appreciate the time and effort that has really contributed to a better manuscript. Here we respond to your questions and hopefully we have made some points more clear.

My main criticism is that as it is presented, the author uses terms that have not yet been explained in enough detail for the reader to understand, and so follow the argument. In particular, line 126: This is the first mention of models or numbers in the main paper. It would be helpful to the reader to define what modelling you are doing and what the numbers mean before referring to them within the main paper. This sentence cannot easily be understood without that extra information. You have not yet stated that you intend to predict PAI with a model before this point in the paper?

Thank you for pointing out that at the moment the manuscript can be confusing for the reader. The Methods section of the manuscript describes in details the models but it is situated after the Discussion section. We have improved the main text to understand what was done and help the reader reconstruct the methods. I demonstrate below how we have done it:

“Repeated TLS data acquired in two transects of 100 x 10 m and one transect of 30 x 10 m between April and October 2019 every 15 days (except between the end of April and early June when the duration between measurements was 40 days) were used to calculate Plant Area Density (PAD, one-sided area of plant material per unit of volume in $\text{m}^2 \text{m}^{-3}$). PAD is a combination of the leaf area and the surface area of woody components, including branches and trunks. An analysis of the vertical profile of the vegetation revealed the existence of only two vertical axes of variation during the dry season, with positive PAD changes below the height of 15 m above the ground (referred to as understory) and negative PAD changes above 15 m height (referred to as upper canopy) (Supplementary Figure 3). The sum of PADs for each 1 m^2 vertical column (X, Y coordinate) were then calculated to obtain Plant Area Index (PAI, one-sided area of plant material per unit of ground surface in $\text{m}^2 \text{m}^{-2}$) (Figure 3a). Non-linear mixed models demonstrated that distance from forest edges has significant effects on PAI within 35 - 40 m of forest margins (Supplementary Figure 2), and we therefore considered edge in this study categorically as the forests within 40 m of the forest fragment margins and interior as the forests at least 40 m distant from the fragment margins. These results demonstrate the existence of vertical and horizontal within-season trends in phenology that should be considered when analysing across-season trends.

The TLS time-series revealed a strong vertical variability in the timing and magnitude of seasonal changes in the PAI of structurally undisturbed forests and forests under edge effects. While transects exhibited similar phenological trends, PAI differed significantly between them (Supplementary Figure 4). We then used linear mixed models to detect the effects of edges on the seasonality of understory PAI, upper canopy PAI, and total PAI (understory + upper canopy PAI), whilst controlling for spatial effects caused by transect differences by including edge effects nested within transect identity as random effects. The most parsimonious model (based on AIC; see Supplementary Table 1) to predict PAI for both the understory and total PAI was equation (1) which includes the additive effects of season and edge effects on PAI, both as categorical variables, and their interactive effects. Equation (2) was selected for upper canopy PAI, which includes the effects of edge and an interaction term between edge effects and season. (Supplementary Table 1; Figure 3; Supplementary Figure 5).”

I also got a bit confused as to which graphs showed observations estimated from the TLS and which showed outputs from your TLS and climate data models? In particular; are figure 3 modelled or measured results? If modelled, as said in the caption, why did you show the model outputs rather than show the measurements you took with the TLS? Or are you referring to TLS to PAI voxel model? You also use a model to relate to climate. Are the two models the same? Please make it clear which model is being used where, and if you did use a model with the climate data, why did you chose to do that rather than analyse the PAD/I estimated from the TLS. A bit more clarification here would be helpful.

Thank you for your questions and for pointing out the lack of clarity. Figure 3 represents the predicted Plant Area Index (PAI) time-series. The reason we presented the predicted values is that we can demonstrate the effects of season and edge effects without taking into account the spatial effects caused by intrinsic characteristics of each transect. Thus, we used mixed linear models that tested for the effects of edge and season (as fixed variables), and included edge effects nested within transect identity as random effects.

Here we present our rationale behind the use of predicted values as opposed to observed values: during the exploration phase of the data, we observed that the transects had similar phenological patterns but differed greatly in their initial PAI. Using mixed modelling helped us control and visualise the combined seasonal and edge effects on canopy PAI. Here we illustrate the difficulties in understanding the phenological patterns that arise from transect-related PAI differences and include the figure in the Supplementary Material (Supplementary Figure 4).

Supplementary Figure 4. Observed Plant Area Index (PAI, $m^2 m^{-2}$) time-series per transect in interior forests (at least 40 m away from forest edges) in Central Amazonia. Large spatial variation in PAI may be attributed to varying local environmental conditions and species composition. While transects exhibited similar (a) understory and (b) upper canopy phenological trends, PAI differed significantly between them. Each point denotes the mean PAI value per transect in each measurement time (with lines denoting linear interpolations between points and colours representing the transect identity). Error bars are the 95% confidence intervals.

I think that it is important to clarify that point above before this paper can be published.

Is it true to say that the total PAI is seasonally invariant? Looking at figure 3, if you add the overstorey and understory there would be a large spike in May, with a much higher PAI than all other months. You show this in supplementary figure 4. What tests did you use to see if there was a seasonal variation?

You are completely right. Our total PAI data were not right and led us to interpret the total PAI trends. Sincere apologies for this mistake and thank you for drawing the attention to the total

canopy PAI trends. The temporal patterns of total PAI in forest edges and forest interior had more complex patterns that reflected the combination of the stratified phenological trends. We have changed figure 3 to include total PAI and illustrate these changes. We also have an additional paragraph in the Results section explaining the total PAI trends:

“The temporal patterns of total PAI in forest edges and forest interior had more complex patterns that reflected the combination of the stratified phenological trends. In the forest interior, a decrease of 2.7% (-0.34 m² m⁻²) was observed between April and early July ($t = -2.8$; P value < 0.005), and remained relatively stable throughout the dry season, except for a peak of plant growth in early September followed by PAI loss in late September. The phenology of forest edges showed very similar trends to forest interior when distinct strata were not considered; the total PAI also decreased by 3.2% (-0.25 m² m⁻²) between April and early July ($t = -2.2$; P value = 0.03), and remained stable throughout the dry season, except for some significant increases of PAI early September followed by PAI decreases in late September. These results show that when the seasonal patterns are not vertically stratified, the PAI trends for the edges versus interiors are strikingly similar and mainly driven by the understory PAI, where the majority of the plant area is.”

Figure 3. Predicted relative Plant Area Index (PAI, %) time-series.

We would like to highlight that these new results, however, do not undermine our conclusions that the understanding of plant phenology of Amazonian forests can be difficult if not impossible to accomplish via passive spaceborne observations. We have included one sentence in the Discussion that highlights the challenges faced by passive sensor to investigate phenological patterns across human-modified landscapes.

“The lack of edge effects on the seasonal variance of total plant area highlights the challenge faced by passive sensors onboard satellite platforms; these systems may suffer from a flattened perspective with data strongly influenced by canopy layers with a denser plant area and little ability to detect significant height-stratified forest canopy responses to climate.”

On a similar subject: Line 129: "Note that the lack of time effect in Eq. 4 indicated that there is significant temporal variation in only the vertical distribution of PAI " - from the supplementary material, figure 4 shows that there is a change over time. Do the error bars here show uncertainty

in the model or spatial variation? If the latter, I do not think it is correct to use that spread as a measure of the significance of the change over time. Please clarify?

Thanks for the comment. The error bars are 95% confidence intervals and reflect the uncertainty in variation of the mean. We have clarified it in the manuscript.

Line 287: You have shown a link between edges and leaf loss, but does this necessarily translate into a carbon loss? Could you provide either data or a citation linking a reduction in mean PAI during the wet season with carbon loss to support that claim?

Thank you for the comment. The links between leaf loss and gross primary productivity (GPP) can be complex and difficult to determine indeed. However, it is already known that plant phenology can affect carbon dioxide fluxes in Amazonian forests (Wu et al., 2016). The total leaf quantity (LAI) is a key component modulating tree growth (Lambers and Oliveira, 2019) and net primary productivity (Reich, 2012). If total CO₂ uptake is suppressed due to reductions in LAI, it may also have negative consequences for the investment in tissue maintenance and defence (Doughty et al., 2015), which may, in turn, increase the mortality of large trees that dominate these upper canopies with large contributions to a reduction in aboveground biomass of these forests. We have included these explanations in the Discussion in the 7th paragraph of the manuscript as follows.

“The total leaf quantity (LAI) is a key component modulating tree growth (Lambers and Oliveira, 2019) and net primary productivity (Reich, 2012). The consistently higher and aseasonal understory PAI in these fragment edges may explain the increased growth rates of understory tree species in these edges in comparison to the same tree species in interior forests (Albiero-Junior, 2021). However, the dry season losses in upper canopy plant area near the edges three months earlier compared with interior forests likely represent a shortening of the photosynthetic-active period of large trees, potentially reducing photosynthetic carbon fixation (gross primary productivity; GPP). If CO₂ uptake of the upper canopy is suppressed, this may have negative consequences for investment in tissue maintenance and defence (Doughty et al., 2015), which may, in turn, increase the mortality of large trees that dominate upper canopies and

contribute to a large reduction in the aboveground biomass of these forests (San-José et al., 2021).”

Figure 4's line fit for the edges seems to be dominated by the one outlier on the far right. Without that one point it looks as if there would have been a negative trend (although not as significant as the overstorey trend). Is that one point certain, or could there have been a mistake or significant occlusion affecting it?

Thank you for your interesting question. Reviewer 1 made a similar comment to which we also gave a detailed response. We have included more information about the graph in the caption. Each black dot represents the mean from 3480 understory and 3480 upper canopy PAI values measured in each survey in the forest interior and 1653 understory and 1653 upper canopy PAI values measured in each survey in the forest edges. The measurements protocol (i.e. position of measurement, LiDAR specifications) remained constant during all periods to minimise analytical errors. To minimise the effect of uncertainties due to occlusion in these dense Amazonian forests, the distance between scanning positions was smaller than the 10-40 m usually applied in previous studies to minimize data uncertainties and maximize data acquisition in the upper canopy. Therefore, we believe that the understory versus upper canopy PAI relationship in the edges is not driven by the one outlier on the far right - as we controlled for sources of uncertainties in the PAI estimation and used a large dataset to obtain the mean value for each period of measurement.

We reran the understory versus upper canopy PAI edge regression without the point in question, finding that the covariation remained weak and not significant:

Figure 4. Observed seasonal changes in upper canopy PAI versus understory PAI. LiDAR-based Plant Area Index (PAI, $m^2 m^{-2}$) between April and October 2019 from Central Amazonian forests. Black dots represent the mean from 3480 understory and 3480 upper canopy PAI values measured in each survey in the forest interior and 1653 understory and 1653 upper canopy PAI values measured in each survey in the forest edges. The highlighted point in red in panel (a) denotes the first TLS measurement made in April 2019. We excluded this point to further investigate the covariance between strata but found no significant relationship (see Supplementary Figure 8). The red lines represent values predicted by simple linear regression (i.e. Understory PAI = $\beta_0 + \beta_1$ Upper canopy PAI), with the shaded grey area depicting the 95% confidence intervals.

We have included the figure below in the Supplementary Material, considering that your question is of great importance and may lead other readers to assume that there is a potentially significant relationship between strata when the point in question is removed from analysis.

Supplementary Figure 8. Observed seasonal changes in upper canopy LiDAR-based Plant Area Index (PAI, $m^2 m^{-2}$) versus understory PAI ($m^2 m^{-2}$) for forest edges without considering the PAI measurements made in April 2019. Black dots represent the mean from 1653 understory and 1653 upper canopy PAI values measured in each survey in the forest edges. The red lines represent values predicted by simple linear regression (i.e. Understory PAI = $\beta_0 + \beta_1$ Upper canopy PAI), with the shaded grey area depicting the 95% confidence intervals.

Line 214 onwards is an interesting discussion, but is it directly relevant to the experiment here and is it supported by the data? If so I think it might be better in the literature review rather than within the discussion?

Thank you very much for your suggestion and we agree that the first sentence of the paragraph (between lines 214 and 2017) is more relevant for the introduction. The remaining part of the paragraph is more relevant to our discussion as we demonstrate how PAI of Amazonian forests follows a seasonal trend, which is aligned to previous findings of a greening but challenges the paradigm that tall trees are limited by sunlight.

I think that supplementary material section 2 needs some more detail. Why are you using a model to relate distance from edge to PAI (or is the model a line fit to suppress noise)? Why not show the raw data? If you do need to use a model, I think you need to justify why it is needed and show its accuracy. That you are using a model suggests that the relationship is very noisy? If so, how significant is the relationship?

Thank you for your question. We have now included the measured PAI data (points) as a function of “distance from edge” in meters and one line indicating the predicted PAI in relation to distance from edge. Variations in PAI due to transect and seasonality are small in comparison to the edge effects on PAI. The results from this model indicates that the effects of transect accounts for 6.5% of the total PAI variability only, and that most of the variance (93.5%) is explained by the within-transect variability, including the distance from edge. Please note that this relationship uses data from the first TLS campaign to investigate

Supplementary section 4 needs more detail for a reader to fully understand and appreciate your points.

Thank you for your comment. We have included some of the description from the Methods in the Supplementary Material. We have specifically included the following text:

“We used a linear mixed-effects (LME) model of understory PAI, upper canopy PAI and a combination of both strata (total PAI) measured from TLS in each transect as a function of time of measurement (*time*) as a categorical variable. We also included an interaction term with the plot category of location near an edge or in the forest fragment interior (*edge effects*) following Qie *et al.* (2017). The *time* × *edge effects* interaction represents how edge effects caused by forest fragmentation influence the seasonal variation in PAI. We compared this LME model with other LME models that contained the variables *time* and *edge effects* as additive terms to examine the significance of seasonality and fragmentation on PAI variation. Model explanatory power was assessed in terms of AIC (Supplementary Table 1). The LME model was fitted using the *lme* function in the *nlme* R package. Variations in transect area and monitoring period can influence PAI trends, and thus we used *varIdent* weights function to account for the noise attributed to sampling effort (Zuur *et al.*, 2009).”

Supplementary materials line 82: "Each point represents the mean values predicted by mixed modelling, with the error bars depicting the bootstrapped 95% confidence intervals." - I do not understand why you are using a model here? You have a direct measure of PAI from the TLS. Why not investigate that rather than passing it through a model, adding in assumptions and errors? A bit more clarification on exactly what analysis you did would be helpful.

Thanks for your question. This question is related to your comment on Figure 3 and we have explained in more detailed the reasons we showed the predicted values instead. In the Supplementary Figures 7 and 7, we also showed the predicted PAI to control for the effects of transects. Thus, we also used mixed linear models that tested for the effects of edge and season (as fixed categorical variables).

REFERENCES

Albiero-Junior, A., Venegas-Gonzalez, A., Camargo, J.L.C., Roig, F.A. and Tomazello-Filho, M., 2021. Amazon forest fragmentation and edge effects temporarily favored understory and midstory tree growth. *Trees*, pp.1-10.

Doughty, C.E., Metcalfe, D.B., Girardin, C.A.J., Amezquita, F.F., Cabrera, D.G., Huasco, W.H., Silva-Espejo, J.E., Araujo-Murakami, A., Da Costa, M.C., Rocha, W. and Feldpausch, T.R., 2015. Drought impact on forest carbon dynamics and fluxes in Amazonia. *Nature*, 519(7541), pp.78-82.

Lambers, H. and Oliveira, R.S., 2019. Plant water relations. In *Plant physiological ecology* (pp. 187-263). Springer, Cham.

Qie, L., Lewis, S.L., Sullivan, M.J., Lopez-Gonzalez, G., Pickavance, G.C., Sunderland, T., Ashton, P., Hubau, W., Salim, K.A., Aiba, S.I. and Banin, L.F., 2017. Long-term carbon sink in Borneo's forests halted by drought and vulnerable to edge effects. *Nature communications*, 8(1), pp.1-11.

Reich, P.B., 2012. Key canopy traits drive forest productivity. *Proceedings of the Royal Society B: Biological Sciences*, 279(1736), pp.2128-2134.

San-José, M., Werden, L., Peterson, C.J., Oviedo-Brenes, F. and Zahawi, R.A., 2021. Large tree mortality leads to major aboveground biomass decline in a tropical forest reserve. *Oecologia*, pp.1-12.

Vincent, G., Antin, C., Laurans, M., Heurtebize, J., Durrieu, S., Lavalley, C. and Dauzat, J., 2017. Mapping plant area index of tropical evergreen forest by airborne laser scanning. A cross-validation study using LAI2200 optical sensor. *Remote Sensing of Environment*, 198, pp.254-266.

Wu, J., Albert, L.P., Lopes, A.P., Restrepo-Coupe, N., Hayek, M., Wiedemann, K.T., Guan, K., Stark, S.C., Christoffersen, B., Prohaska, N. and Tavares, J.V., 2016. Leaf development and demography explain photosynthetic seasonality in Amazon evergreen forests. *Science*, 351(6276), pp.972-976.

Zuur, A.F., Ieno, E.N., Walker, N.J., Saveliev, A.A. and Smith, G.M., 2009. *Mixed effects models and extensions in ecology with R* (Vol. 574). New York: Springer.

Reviewer #3 (Remarks to the Author):

Overall comment:

In studying the influence of fragmentation on tropical forest phenology and density of vertical biomass, the authors investigate a potentially significant phenomenon with implications for biome-wide estimates of ecosystem function. The topic is of particular concern given the prevalence of forest edges across the Amazon.

The authors examine seasonal changes in the vertical biomass profile (i.e. understory vs. overstory) of the canopy, which as they point out can be difficult if not impossible to accomplish via passive spaceborne observations. This is due to the fact that the green biomass signal is integrated vertically and therefore cannot be decomposed into understory vs. overstory components in passive remote sensing data. The authors utilize terrestrial laser scanning in order to resolve these two canopy components through a series of field observations. The dataset is impressive with observations collected every 15 days to capture within- and across-season trends in phenology.

As a scientist with much experience in terrestrial laser scanning I find the remote sensing methods (data collection, processing, and analysis) to be solid. The results show small yet statistically significant trends in the PAI data; of particular note is the low level of PAI variation (due to the fact that ~90-97% of the canopy is invariant through time—comprised of woody and non-deciduous foliar elements). While this might be seen as a limitation to this study (i.e. that the lidar data do not allow resolution of foliar elements as well as some passive spectral approaches), I find the large size and novelty of the dataset in characterizing detailed PAI data is quite attractive and represents an important advance in our understanding of tropical forest phenology relative to disturbance.

Thank you very much for appreciating our effort in collecting and processing the large amount of repeat terrestrial laser scanning data. We are enthusiastic about our findings and, as you mentioned, we also think that this work represents an advance in the understanding of phenology relative to disturbance and challenges some of the paradigms that Amazonian forests are limited by light, as we demonstrate that temperature (and possibly VPD) may play an important role in

the canopy dynamics. Thank you very much for your time to provide detailed comments. They were crucial to improving the quality of the paper - especially on the total PAI section as described below. We hope that we have replied to your comments accordingly.

Detailed Comments:

Lines 85-88: It would add clarity to the paper and help with understanding the specific scientific goals of the paper if the authors could frame their objectives using hypotheses. This is particularly necessary because three times later in the paper the authors allude to hypotheses (lines 182, 422, and 436) that were never clearly stated, and this is the place to do it.

Thank you for highlighting this point. We have made our hypothesis more clear.

“we hypothesised that: (1) vertically stratified plant phenology in undisturbed forests varies with seasonal microclimatic conditions; (2) the understory phenology is dependent on seasonal variations in the upper layers of the canopy; and (3) plant phenology is sensitive to disturbances arising from forest fragmentation, with the hotter and drier conditions of edges exacerbating leaf and branch losses during the dry season”.

Figure 1: Please edit caption to say “...depicting plants from distinct vertical strata.” Also to help the reader interpret the scan image, please include a scale bar indicating the height of vegetation relative to the colors shown on the graphic.

Thank you! We have accepted your suggestions and included a scale bar in Figure 1.

Figure 2, line 118: Suggest: “...Fifth order polynomial models were fit to the microclimate data for visualization purposes...”

Thank you for your comment. We have accepted your suggestion.

Equations, lines 132-133: Unless I'm missing a formatting protocol regarding the Methods, these should be labeled Eq.'s 1 and 2.

According to Nature Communication's formatting protocol, “Equations that are referred to in the text are identified by parenthetical numbers, such as (1), and are referred to in the manuscript as "equation (1)".” We will keep this format throughout the manuscript.

Figure 3: This is an interesting figure however I feel that it can be much improved. I suggest that the authors use a **consistent scale bar on the y-axis** so that the trends can be compared and contrasted more readily (in fact plotting upper/lower canopy on one graph each would likely be even better). In addition, to assist with interpretation of the overall forest trends, I suggest that a third series of plots be added that shows the overall PAI of the forest canopy in the forest edges vs. forest interior. This can be added without adding space to the figure, as in my opinion the figure would be improved if there was only one graph each for edges and interior forest locations, with each graph having three lines (upper, lower, and total canopy).

Thank you for your comment. As recommended, we have condensed to 2 graphs (Figure 3b and 3c) to better visualise the phenological patterns in the edge and interior and included lines for upper canopy, understory, and total PAI in each panel. We have transformed the values into relative PAI (%) so that comparisons are easier. However, we believe that showing the data in PAI ($\text{m}^2 \text{m}^{-2}$) is also relevant as it gives a better picture of the magnitude of changes. These graphs are presented in the Supplementary Material as Supplementary Figure 5. Each point in the graph represents the predicted PAI from a subset of the data (more information is included in the figure caption). We also include the Figure 3a that help visualise the edge versus interior, as well as understory versus upper canopy thresholds, with absolute PAI values in April 2019 predicted from linear mixed models.

Figure 3. Predicted relative Plant Area Index (PAI, %) time-series. PAI predictions from linear mixed modelling used date of LiDAR measurements and the interaction with a categorical variable indicating whether plots were near an edge as fixed variables. Edge effects nested within transect identity were included as random variables (equations 1 and 2). Predicted PAI of the understory (< 15 m canopy height), upper canopy (≥ 15 m canopy height) and total PAI that combined both vertical strata in b) forest edges and c) undisturbed interior forests. Forest edges are defined as canopies within 40 m from forest margins while forest interior are canopies at least 40 m away from the forest fragment margins. Relative PAI was calculated as the PAI at any time divided by the initial PAI. Each point (and lines denoting linear interpolations between points) represents the mean relative PAI obtained by fitting 200 randomised permutations of

subsets split into 80/20 for calibration and validation, respectively. The shaded areas represent 95% confidence intervals based on uncertainty in those parameter estimates. While transects exhibited similar phenological trends, PAI differed significantly between transects (Supplementary Figure 4 for measured PAI). For absolute predicted PAI values and model uncertainty, see Supplementary Figure 5. The shaded grey area represents the dry season.

Line 178: I don't believe you have leaf area data? Only plant area index, of which a very large component (~90-97%) apparently is comprised of non-deciduous plant material based on Figure 3.

Unfortunately we do not have leaf area data. Indeed, automated methods that separate leaves from woody materials do exist (i.e. [1] Vicari *et al.*, 2019; [2] Wang *et al.*, 2020). However, the high structural complexity of the Amazonian forests where our measurements were made (including the high abundance and diversity of lianas that are higher in close proximity to these forest edges, [3] Laurance *et al.*, 2001), pose a challenge to the separation of leaves and may contribute to additional uncertainties that can vary through space (edge versus interior) and time (species-specific phenological trends that vary with time of year). Considering the small changes in PAI across seasons, the varying uncertainties could lead to a fallacy of interpretation.

Lines 223-224: Showing these total canopy PAI data would be helpful (i.e. as a third line on Figure 3) so that the reader can follow. Also--It would seem that since the upper/lower canopy PAI trend is not negatively correlated in the disturbed edge effect case, that in fact there may in this case be a total canopy PAI sensitivity to climatic seasonality—please explain.

Thank you very much for this observation - you are right! First of all, the total PAI values presented in the Supplementary Material were wrong. Sincere apologies for this mistake and thank you for drawing the attention to the total canopy PAI trends. The phenology patterns from total PAI in edges and forest interior have more complex patterns that reflected the combination of the stratified phenological trends. We have an additional paragraph in the Results section explaining the total PAI trends:

“The temporal patterns of total PAI in forest edges and forest interior had more complex patterns that reflected the combination of the stratified phenological trends. In the forest interior, a decrease of 2.7% ($-0.34 \text{ m}^2 \text{ m}^{-2}$) was observed between April and early July ($t = -2.8$; P value <

0.005), and remained relatively stable throughout the dry season, except for a peak of plant growth in early September followed by PAI loss in late September. The phenology of forest edges showed very similar trends to forest interior when distinct strata were not considered; the total PAI also decreased by 3.2% ($-0.25 \text{ m}^2 \text{ m}^{-2}$) between April and early July ($t = -2.2$; P value = 0.03), and remained stable throughout the dry season, except for some significant increases of PAI early September followed by PAI decreases in late September. These results show that when the seasonal patterns are not vertically stratified, the PAI trends for the edges versus interiors are strikingly similar and mainly driven by the understory PAI, where the majority of the plant area is.

”

We would like to highlight that these new results, however, do not undermine our conclusions that the understanding of plant phenology of Amazonian forests can be difficult if not impossible to accomplish via passive spaceborne observations. We have included the following text in the Discussion:

“The lack of edge effects on the seasonal variance of total plant area highlights the challenge faced by passive sensors onboard satellite platforms; these systems may suffer from a flattened perspective with data strongly influenced by canopy layers with a denser plant area and little ability to detect significant height-stratified forest canopy responses to climate.”

Line 307: I suggest that you use an actual spatial indicator of scale here (e.g. “Landscape”, “Regional”, “Continental” rather than “satellite” scale, which is vague since satellite observations occur from local to global scales.

Thanks for your comment and we agree that this was vague. We have changed it to “landscape and regional scales”.

Line 393: “as follows:”

Corrected.

Line 397: This first sentence is duplicative as it has already been stated that voxels are one cubic meter in volume.

Thank you and we agree. We have removed the sentence.

Lines 408-419: This section would be better placed in the Discussion section, in my opinion.

It is true! Thank you for your comment and we have included it as the 5th paragraph of the Discussion.

REFERENCES

Laurance, W.F., Pérez-Salicrup, D., Delamônica, P., Fearnside, P.M., D'Angelo, S., Jerozolinski, A., Pohl, L. and Lovejoy, T.E., 2001. Rain forest fragmentation and the structure of Amazonian liana communities. *Ecology*, 82(1), pp.105-116.

Vicari, M.B., Disney, M., Wilkes, P., Burt, A., Calders, K. and Woodgate, W., 2019. Leaf and wood classification framework for terrestrial LiDAR point clouds. *Methods in Ecology and Evolution*, 10(5), pp.680-694.

Wang, D., Momo Takoudjou, S. and Casella, E., 2020. LeWoS: A universal leaf- wood classification method to facilitate the 3D modelling of large tropical trees using terrestrial LiDAR. *Methods in Ecology and Evolution*, 11(3), pp.376-389.

Reviewer comments, second round review:

Reviewer #1 (Remarks to the Author):

I appreciate that the authors changed the language describing a possible connection between the phenology changes and "forest loss". I think the new language is more accurate, although there are several steps of assumptions to get from phenological change to "forest loss" (phenological change -> decreased photosynthesis -> decreased CO₂ uptake -> decreased investment in plant defense -> increased mortality of large trees). This takes some of the punch out of the importance of the findings in my view, appropriately. I do not mean to diminish the accomplishments of the study, but this is a long way from the original wording that the phenological change will cause "forest loss".

I also appreciate the authors acknowledgement of the need for caution in interpreting these findings generally, and I think the wording is more accurate on this now too. The authors refer to other studies also with limited scope, nonetheless the inference must match the study. The other reviewers seem to be more impressed by the dataset, so maybe I'm being too stringent on that.

Fig 4: perhaps leave out the solid line in (a) (or make dashed); it's confusing when it's not a significant relationship. The removal of the outlier did switch the direction of the trend.

Reviewer #2 (Remarks to the Author):

The comments from the first round have been addressed.

Reviewer #3 (Remarks to the Author):

I am satisfied with the revisions made to the manuscript. It was clear that the authors were diligent and sincere in their responses to my comments, questions, concerns, and suggestions. I am in less of a position to comment specifically to the points of the other reviewers, however in my read of the response manuscript I feel that the authors did a good job and that the manuscript is now much improved. In my opinion this paper should be a strong contribution. Thank you.